# The Modeling of a River Impacted with Tailings Mudflows Based on the Differentiation of Spatiotemporal Domains and Assessment of Water–Sediment Interactions Using Machine Learning Approaches



João Paulo Moura [1], Fernando António Leal Pacheco [2,*], Renato Farias do Valle Junior [3], Maytê Maria Abreu Pires de Melo Silva [3], Teresa Cristina Tarlé Pissarra [4], Marília Carvalho de Melo [5], Carlos Alberto Valera [6], Luís Filipe Sanches Fernandes [1] and Glauco de Souza Rolim [4]

[1] CITAB—Centro de Investigação e Tecnologias Agroambientais e Biológicas, Universidade de Trás-os-Montes e Alto Douro, Ap. 1013, 5001-801 Vila Real, Portugal; jpmoura@utad.pt (J.P.M.); lfilipe@utad.pt (L.F.S.F.)

[2] CQVR—Centro de Química de Vila Real, Universidade de Trás-os-Montes e Alto Douro, Ap. 1013, 5001-801 Vila Real, Portugal

[3] Instituto Federal do Triângulo Mineiro, Campus Uberaba, Laboratório de Geoprossessamento, Uberaba 38064-790, MG, Brazil; renato@iftm.edu.br (R.F.d.V.J.); mayte@iftm.edu.br (M.M.A.P.d.M.S.)

[4] Faculdade de Ciências Agrárias e Veterinárias, Universidade Estadual Paulista (UNESP), Via de Acesso Prof. Paulo Donato Castellane, s/n, Jaboticabal 14884-900, SP, Brazil; teresa.pissarra@unesp.br (T.C.T.P.); glauco.rolim@unesp.br (G.d.S.R.)

[5] Secretaria de Estado de Meio Ambiente e Desenvolvimento Sustentável, Cidade Administrativa do Estado de Minas Gerais, Rodovia João Paulo II, 4143, Bairro Serra Verde, Belo Horizonte 31630-900, MG, Brazil; marilia.melo@meioambiente.mg.gov.br

[6] Regional Coordination of Environmental Justice Promoters of the Paranaíba and Baixo Rio Grande River Basins, Rua Coronel Antônio Rios, 951, Uberaba 38061-150, MG, Brazil; carlosvalera@mpmg.mp.br

* Correspondence: fpacheco@utad.pt

**Abstract:** The modeling of metal concentrations in large rivers is complex because the contributing factors are numerous, namely, the variation in metal sources across spatiotemporal domains. By considering both domains, this study modeled metal concentrations derived from the interaction of river water and sediments of contrasting grain size and chemical composition, in regions of contrasting seasonal precipitation. Statistical methods assessed the processes of metal partitioning and transport, while artificial intelligence methods structured the dataset to predict the evolution of metal concentrations as a function of environmental changes. The methodology was applied to the Paraopeba River (Brazil), divided into sectors of coarse aluminum-rich natural sediments and sectors enriched in fine iron- and manganese-rich mine tailings, after the collapse of the B1 dam in Brumadinho, with 85–90% rainfall occurring from October to March. The prediction capacity of the random forest regressor was large for aluminum, iron and manganese concentrations, with average precision > 90% and accuracy < 0.2.

**Keywords:** river; spatiotemporal domain; sediment source; metals; machine learning prediction

## 1. Introduction

Rivers are primary pathways for water and sediment transport and environments of water–sediment interactions. However, understanding and predicting the water composition responses to these interactions is challenging, because the interactions are subject to multiple factors that can change them over time and space [1]. One important factor is sediment change downstream, particularly during mixing with inflows from tributaries or hillslopes [2–4]. Rainfall runoff variations, either in the spatial or the temporal domains, can also change the water–sediment interactions, as they affect sediment delivery to streams [5,6], cause dilution effects [7], or trigger sediment resuspension during storm

events [8,9]. A third important factor consists in changes of water temperature, pH or redox conditions in the water column, as they affect the mobility of elements, especially of metals [10,11]. Thus, the modelling of water–sediment interactions in rivers must consider proper spatial and temporal domains where sediment sources, the hydrologic regime and physical–chemical parameters of river water, are relatively homogeneous.

Having defined the proper spatial and temporal frames for the water–sediment interaction model, the next step is to select a method to resolve it. A response to this request can be hydrological [12,13] or based on data analysis [14,15]. Hydrological models allow unraveling the processes of flow (including extreme flows and floods), water and sediment transport and ecosystem functioning, among many other issues related with riverine systems, and have been recently developed and used in various studies [16–22]. In this regard, it is worth acknowledging the growth in open source software or scripts [23,24], which makes hydrologic modeling fair [25]. However, in general, these models are challenging with regard to data availability and calibration in similar work [26–29], especially in large and heterogeneous river basins, despite the attempts to handle these issues effectively [30]. In these cases, data analysis methods may be the solution to understand and predict the composition of river water. Conventional statistics (univariate, bivariate and multivariate) enables establishing a comprehensive picture of relationships between element concentrations and their determinant factors (e.g., streamflow, size of sediment particles, water pH or temperature), while artificial intelligence algorithms may be sufficiently precise and accurate to allow reliable predictions of element concentrations in the near future.

In the most recent years, the use of artificial intelligence algorithms has intensified in the modelling of streamflow [31] and sediment transport [32–37]. However, there are few or no studies that have used these algorithms for the modelling of water–sediment interactions. Even less common are those studies that have coupled statistical with artificial intelligence models to simultaneously describe how the water composition responds to interactions with sediments and the environment and provide water composition forecasts within proper spatial and temporal domains.

In order to help reduce the scarcity of studies capable of predicting element concentrations resulting from water–sediment interactions based on artificial intelligence methods, the general purpose of this study was to develop a river model that describes the interaction between the water compartment (physical and chemical characteristics) and the sediment compartment (granulometric and chemical characteristics), while also considering the influence of sediment source and streamflow. The pathway to achieve the general objective comprised splitting the workflow into two phases. The first phase was called the experimental phase, and encompassed data analyses using conventional statistical methods (e.g., boxplots, Spearman correlation coefficients, principal component analysis). This phase allowed for understanding of the dynamics of metals and their changes/interactions in the water and sediment compartments, both in the spatial domain, i.e., considering areas of a river with different characteristics, and in the temporal domain, i.e., distinguishing dry from rainy periods. In the second phase, called the modeling phase, the concentrations of various metals (e.g., aluminum, iron and manganese) were modelled in the studied area using artificial intelligence algorithms, separately for each spatiotemporal domain. Several algorithms of different nature and complexity were tested, with the aim of evaluating and comparing the precision and accuracy of the generated models. In the present study, the selected algorithms were: multiple linear regression with stepwise forward selection of variables, multilayer perceptron neural networks and random forest regressor, using dissolved and total element concentrations as targets (dependent variables), and tens of physical and chemical characteristics of water and sediment as features (independent variables).

The river model was tested in the Paraopeba River, located in the state of Minas Gerais, Brazil. This river was selected because of its marked segmentation in the spatial and temporal domains, which occurred after the rupture of a mine-tailings dam (B1) on 25 January 2019 in a tributary stream called Ribeirão Ferro-Carvão. The accident has divided the Paraopeba River into three main segments: (1) the "upstream" segment, not affected

by the accident and where the streamflow carries natural sediments; (2) the "anomalous" segment, where the tailings overlayed the natural sediments and the two fractions have been dislocated downstream since then; (3) the "natural" segment, located far from the accident, and where the streamflow also carries natural sediments, but not necessarily equal to segment 1 sediments, considering the inflows of tributaries that occur between the two regions. Besides the current spatial segmentation, the Paraopeba River basin is located in a region of tropical climate, where rainfall changes greatly in the transition between dry and the rainy seasons, with 85–90% of all rainfall occurring in the latter period. This climate framing renders the possibility to investigate water–sediment interactions under well-defined temporal domains, namely, those of low and high river flows.

The B1 dam was located in the municipality of Brumadinho and the rupture spilled about 11.7 $Mm^3$ of iron- and manganese-rich mine tailings into the Ribeirão Ferro-Carvão watershed, of which 2.8 $Mm^3$ reached the Paraopeba River some 10 km downstream [38]. The accident had immediate management consequences, such as the prohibition of drinking water supply from the impacted areas of the Paraopeba River [39] due to the high levels of heavy metals dissolved in the water, namely, aluminum (Al), iron (Fe) and manganese (Mn). Among other urban centers, this ban affected the supply to the Metropolitan Region of Belo Horizonte with a population of about 6 million inhabitants [40]. In addition to the aforementioned ban, a monitoring plan of metal concentrations in sediments and water of the Paraopeba River was implemented, with high spatial and temporal resolution, namely, one monitoring station every 15 km along the river, from the confluence with the Ribeirão Ferro-Carvão to the mouth of the Paraopeba River at the junction with the São Francisco River, and daily or weekly measurements from 2019 to the present [41,42].

The experimental phase of our study, which, as indicated above, used conventional statistical methods to assess water–sediment or water–tailings interactions, is expected to bring knowledge on the relevance of countless data collected and monitored along the Paraopeba River. That knowledge is important to make the best use of artificial intelligence algorithms to obtain realistic river models of water–sediment and water–tailings interactions. The machine learning neural networks trained for the Paraopeba River, where a technological accident occurred with the release of various contaminants (metals), will define predictive models that will allow, in future works: (i) to estimate the time needed to bring the current contamination scenario towards pre-rupture conditions; (ii) to measure the spatiotemporal impact of eventual preventive or corrective actions to be applied in the Paraopeba River basin. Taking all these insights and potential actions together, the main contribution and novelty of this work lies in combining, in a single study, the segmentation of a river system that was divided into three segments due to a technological accident: not impacted, impacted and potentially impacted. Each homogeneous sector was analyzed from a geochemical and environmental point of view, using artificial intelligence approaches capable of linking the immense number and diversity of variables that contribute to the quality of water in rivers and sediments and, most importantly, building a predictive structure capable of anticipating the river's future.

## 2. Materials and Methods

### 2.1. Study Area

The study area comprises the Paraopeba River basin (area: 13,600 $km^2$), located in the Brazilian state of Minas Gerais (Figure 1). The basin rises from the mouth, located at the confluence with the São Francisco River (altitude: 553 m.a.s.l.), where the Três Marias reservoir was created by the Bernardo Mascarenhas hydroelectric dam, to the spring located in the south edge of Espinhaço mountain range, Cristiano Otoni municipality (altitude: 1610 m.a.s.l). The relief has a strong influence on precipitation. In general, the annual rainfall increases from the lowlands to the highlands, ranging from 1185 to 1750 mm·year$^{-1}$. Besides topography, seasonality is also a great factor in rainfall variation, because 85–90% of all precipitation occurs in a rainy period generally running from October to March.

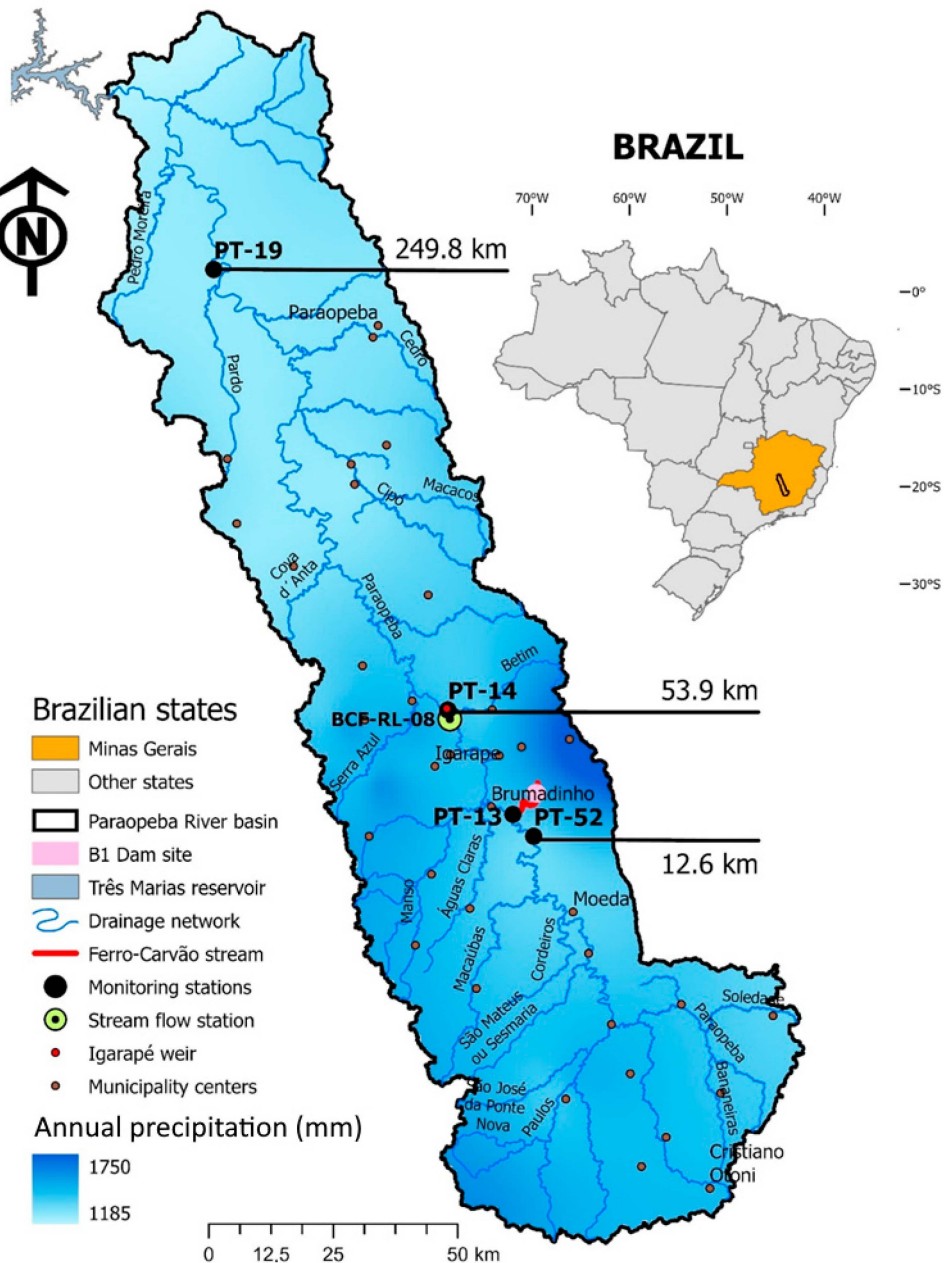

**Figure 1.** Map of Paraopeba River basin, with indication of major tributaries, municipality centers, position in the Brazilian state of Minas Gerais, and location of four water and sediment monitoring stations: PT-52 ("upstream"), PT-13, PT-14 ("anomalous") and PT19 ("natural"), as well as one streamflow station BCF-RL-08, all used in the statistical assessment and artificial intelligence modeling of water–sediment interactions. The shaded areas describe the general precipitation increase from the mouth to the spring areas of the basin. The datasets of streamflow and water and sediment parameters are provided as Supplementary Materials.

In the municipality of Brumadinho, the Vale S.A. company of Rio de Janeiro, Brazil, a major ore extractor in Brazil and worldwide, owned the tailings dam of the Córrego do Feijão mine, known as dam B1. This dam collapsed on 25 January 2019, spilling about 11.7 Mm$^3$ of iron- and manganese-rich tailings into the watershed of Ribeirão Ferro-Carvão, a tributary of the Paraopeba River with an area of about 32.8 km$^2$. The tailings moved along the Ribeirão Ferro-Carvão, first as an avalanche and then as a debris flow and mud flow, and 2.8 Mm$^3$ of tailings ended up entering the Paraopeba River, nearly 10 km downstream, depositing over and mixing with the natural river sediments. In the area closest to dam

B1, the debris resulting from the collapse (tailings + soil ripped from the stream bed) had a particle size distribution characterized by 22.5% sand (very fine to medium), 58.9% silt and 18.5% clay. This torrent lost energy over the sinuous course of Ribeirão Ferro-Carvão so that, when the 2.8 $Mm^3$ of tailings entered the Paraopeba River, they were just composed of very fine fractions, namely, 63.4% silt and 36.3% clay. Regarding the chemical composition, this mud contained aluminum (16.8 $g \cdot kg^{-1}$), iron (139.5 $g \cdot kg^{-1}$) and manganese (7.9 $g \cdot kg^{-1}$), and smaller proportions of phosphorus, arsenic and lead, with an average density of 3.5 $g \cdot cm^{-3}$ [43]. When compared with the natural sediments of the Paraopeba River, the mud was enriched in the finest fractions, because the natural sediments had, on average, 46.9–58.3% silt (dry period–rainy period) and only 2.3–3.9% clay. It was also enriched in iron and manganese, but not in aluminum, as, in the dry period, the natural sediments of the Paraopeba River presented average concentrations of aluminum, iron and manganese of 18.6%, 79.3% and 1.6%, respectively, and in the rainy period, of 20.9%, 68.3% and 1.9%, respectively (refer to the Supplementary Materials).

*2.2. Dataset*

Since the date of the B1 dam disaster, the Vale S.A. company implemented a monitoring plan, comprising daily water and sediment sampling accompanied by the physical–chemical characterization of 270 parameters at 65 different locations [41,42], as ordered by a court decision, to prevent the spill from spreading to the Atlantic Ocean. Although implemented by the company that was responsible for the accident, the monitoring plan is overseen by the Public Prosecution Service of Minas Gerais and the Minas Gerais Institute for Water Management, in the context of a Judicial Agreement [44]. The data were made available for this study in the context of a contract between the Vale, S.A. company and the higher education institutions affiliated with the study, as indicated in the Acknowledgements section.

The Vale S.A. company is required by court order to collect a large amount of data, for example, to determine the presence of hydrocarbons in the water. Based on our experience from several studies already carried out, we selected the data suitable for this study. The values of 30 parameters were compiled from the data records, spanning the January 2019–December 2021 period. The list of parameters is depicted in Table 1. The datasets were organized into 6 compartments. Three of them (A, B, E) contained data about 6 contaminants: aluminum, arsenic, lead, iron, phosphorus and manganese, with the respective concentrations in water (dissolved and total) and in sediment + tailings mixtures. A fourth compartment (C) covered the data related to the river conditions: dissolved oxygen, pH, oxy-reduction potential, temperature and turbidity. A fifth compartment (F) covered the granularity of sediments and tailings: clay, silt, very fine-grained sand, fine-grained sand, sand, coarse-grained sand and very coarse-grained sand. Finally, the flow of the Paraopeba River was integrated in the (D) compartment.

The data on the 30 parameters were compiled from the records of four water and sediment monitoring stations, represented in Figure 1, which were selected among the 65 monitored locations to represent specific domains within the Paraopeba River. The raw data are provided as Supplementary Materials. The measurement protocols are described in Section 2.2.1. After the rupture of dam B1 in Brumadinho, the Paraopeba River was divided into three regions, namely "upstream", "anomalous" and "natural", justified by the disparity of iron and manganese concentrations measured in the sediments or sediment + tailings mixtures. In Figure 1, it is evident that the station designated "upstream" (PT-52) is located 12.6 km upstream of dam B1, meaning in a sector of the Paraopeba River exempt from the effects of the accident. It also shows that the "anomalous" segment is located between of the confluence of Ribeirão Ferro-Carvão with the Paraopeba River and the physical barrier called Igarapé weir (PT-14, located 53.9 km downstream of the B1 dam site). The PT-13 station, hereafter referred to as the "anomalous station", corresponds to the sector most impacted by the tailings dump, as per the report of the Arcadis company about mechanical drillings executed along the river and corresponding characterization of

sediment and tailings testimonies [45]. And finally, the "natural" station (PT-19) is located 249.8 km downstream of dam B1, corresponding to a sector not yet affected by the dam break as per the same report of Arcadis company.

**Table 1.** Compartments and corresponding parameters assessed in the monitoring stations PT-52 ("upstream"), PT-13 ("anomalous") and PT-19 ("natural"), as well as in the BCF-RL-08 streamflow station, used in the statistical and artificial intelligence models. The dataset of weakly averaged values for all these parameters is provided as Supplementary Materials.

| Compartment | | Variable | Description | Unit |
|---|---|---|---|---|
| A | River water chemistry (concentrations of contaminants; dissolved) | Al(dis) | Dissolved Aluminum | mg L$^{-1}$ |
| | | As(dis) | Dissolved Arsenic | mg L$^{-1}$ |
| | | Pb(dis) | Dissolved Lead | mg L$^{-1}$ |
| | | Fe(dis) | Dissolved Iron | mg L$^{-1}$ |
| | | P(dis) | Dissolved Phosphorus | mg L$^{-1}$ |
| | | Mn(dis) | Dissolved Manganese | mg L$^{-1}$ |
| B | River water chemistry (concentrations of contaminants; total) | Al(tot) | Total Aluminum | mg L$^{-1}$ |
| | | As(tot) | Total Arsenic | mg L$^{-1}$ |
| | | Pb(tot) | Total Lead | mg L$^{-1}$ |
| | | Fe(tot) | Total Iron | mg L$^{-1}$ |
| | | P(tot) | Total Phosphorus | mg L$^{-1}$ |
| | | Mn(tot) | Total Manganese | mg L$^{-1}$ |
| C | River water condition | DO | Dissolved Oxygen | mg L$^{-1}$ |
| | | pH | pH | |
| | | Eh | Redox Potential | mV |
| | | T | Temperature | °C |
| | | Tb | Turbidity | NTU |
| D | Streamflow | Q | Streamflow | m$^3$ s$^{-1}$ |
| E | Tailings/sediment chemical composition | Al(sed) | Aluminum | mg L$^{-1}$ |
| | | As(sed) | Arsenic | mg L$^{-1}$ |
| | | Pb(sed) | Lead | mg L$^{-1}$ |
| | | Fe(sed) | Iron | mg L$^{-1}$ |
| | | P(sed) | Phosphorus | mg L$^{-1}$ |
| | | Mn(sed) | Manganese | mg L$^{-1}$ |
| F | Tailings/sediment grainsize fractions | Clay | Clay (0.0002–0.00394 mm) | g kg$^{-1}$ |
| | | Silt | Silt (0.00394–0.062 mm) | g kg$^{-1}$ |
| | | sandVF | Very fine-grained sand (0.062–0.125 mm) | g kg$^{-1}$ |
| | | sandF | Fine-grained sand (0.125–0.25 mm) | g kg$^{-1}$ |
| | | sandM | Sand (0.250–0.500 mm) | g kg$^{-1}$ |
| | | sandC | Coarse-grained sand (0.500–1.000 mm) | g kg$^{-1}$ |
| | | sandVC | Very coarse-grained sand (1.00–2.00 mm) | g kg$^{-1}$ |

The statistical and artificial intelligence models were based on weekly averages of each time series presented in Table 1, smoothed by a moving average process of order 4 as advocated by [46] for time series analysis. The use of a moving average plays a filtering role, minimizing outliers but also attenuating peaks. The main advantage is to reduce the time lag between the response of dependent variables (e.g., the metal concentrations) to the evolution of streamflow. Our main independent variable is the streamflow, and we know that there is always a delay between the moment of its increase or decrease and the corresponding effect on the other variables. Thus, through the moving average process, we minimize the lag phase among the intervening variables.

In addition to the physical–chemical parameters measured in the water and sediment samples, the models considered the flow records measured at the BCF-RL-08 hydrometric station of the Vale, S.A. company (Figure 1). The database used in the statistical assessment and artificial intelligence modeling is presented as Supplementary Materials, where the laboratories and analytical methods used in the determination of each parameter are also mentioned. Considering the large number of variables used in the current study, a list of notations and abbreviations is provided in Abbreviations.

### 2.2.1. Measurement Protocols

Chemical and granulometric analyses of tailings, sediments and water were carried out using industry-standard techniques in the laboratories of SGS and BIOAGRI [47]. The ABNT NBR ISO/IEC 17025:2017 standard qualified both laboratories, ensuring their compliance with traceable quality assurance/quality control (QA/QC) policies and processes [47]. The accreditation codes are CRL-0172 for BIOAGRI and CRL-386 for SGS.

Laser diffraction was used to assess the grain sizes of the tailings and sediments according to ISO 13320:2009, while thermogravimetry (Leco, model 701) and pycnometry (Quantachrome Instruments, model ULTRAPYC 1200e) were used to calculate loss on ignition (LOI) and density, respectively. Inductively coupled plasma atomic emission spectrometry (ICP-AES; Agilent, model 5110) was used for measurement and EPA method 3050B—"Acid digestion of sediments, sludges and soils" was used for sample preparation [47]. The samples were examined using fused pellet analysis, and the amounts of major and minor elements (such as Si, Al, Fe, Ti, Ca, Mg and P) were assessed using an X-ray fluorescence spectrometer (Panalytical, model Axios Minerals) [47]. Vale SA's Mineral Development Center carried out the mineralogical analyses (including oxides of quartz, feldspar, Fe and Mn, among other minerals). QEMSCAN identified the constituent minerals by combining electron microscopy and EDS microanalysis pixel by pixel.

Multiparameter probes were used to measure the physical characteristics of the water in situ. The Standard Method for the Examination of Water and Wastewater no. 3125—"Metals in Water by Inductively Coupled Plasma Mass Spectrometry" (ICP-MS) was used for measurement, and no. 3030—"Nitric Acid Digestion of Metal Samples" was used for sample preparation in the chemical analysis of metals and arsenic in water [47]. The standard technique of the "Ascorbic Acid Method" no. 4500-PE was used to quantify phosphorus [47].

### 2.3. Model Framework

The workflow is illustrated in Figure 2, which generally describes the model framework proposed in this study to assess water–sediment interactions in large rivers. The dataset preparation, based on the definition of compartments, has been described in Section 2.2. From that stage onwards, the river model sets up the working spatial and temporal domains. In general, the spatial domain is thought to represent regions of dominant sediment sources, either natural (e.g., determined by geology) or anthropogenic (e.g., determined by land use). In the present study, the river model embedded a spatial domain composed of three regions, two of them dominated by natural sediments (the "upstream" and "natural" segments) and a third dominated by natural sediments overlaid or mixed with mine tailings (the "anomalous" segment). The temporal domain is meant to discriminate periods of contrasting rainfall or streamflow, which, in the present study, were linked to the dry and rainy periods of Brazil's tropical climate.

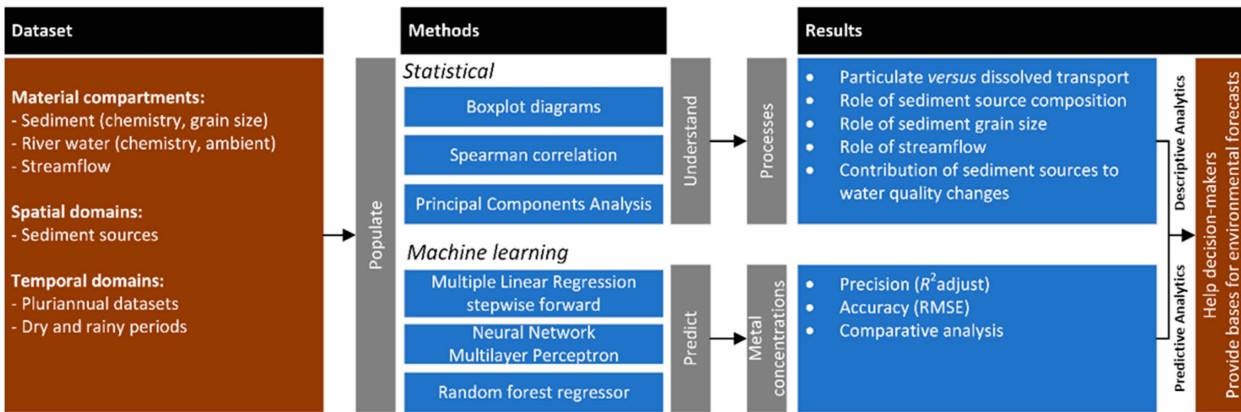

**Figure 2.** River model proposed to assess water–sediment interactions in large rivers, based on the initial definition of spatial and temporal domains succeeded by the application of an ensemble of statistical and artificial intelligence algorithms and integrated interpretation of their results.

### 2.3.1. Experimental Phase

Having defined the dataset as well as the spatial and temporal domains, this information feeds univariate, bivariate and multivariate statistical models, all integrated in what we called the "experimental phase". The observed values of each parameter are obtained by measurement using suitable and properly calibrated sensors. The sensors have their accuracy specifications and confidence intervals, which can be checked in the references listed in Section 2.2.1. Measurement protocols.

In the first step, descriptive statistical analyses are applied to a list of parameters (i.e., Table 1) measured at the monitoring stations that represent the spatial domains, with the purpose of assessing their variability (univariate analysis). The analyses are repeated for the dry and rainy periods to assess seasonally related variability. Complementing the numerical assessments, the univariate analyses comprise the drawing of boxplot diagrams to facilitate interpretation. The descriptive statistical analyses are followed by bivariate and multivariate modeling in a second step. In the present study, we applied these and subsequent analyses only to the "upstream" and "anomalous" PT-13 segments, because they were the most contrasting sectors of the Paraopeba River in terms of their relationship to the B1 dam collapse (non-impacted and impacted, respectively), and hence the best to differentiate water–sediment interactions according to sediment source (natural sediments or natural sediments + mine tailings, respectively). Thus, to understand the relationship between the measured variables, Spearman rank-order correlation and principal component analysis were performed. Spearman correlation coefficients are computed to capture the magnitude of bivariate relationships among the relevant variables (e.g., those listed in Table 1), and principal component analysis is used to identify the most relevant variables in the system.

### 2.3.2. Modelling Phase

In the second phase of the workflow depicted in Figure 2, known as the "modeling phase", artificial intelligence models were trained. The choice of machine learning models is due to the fact that they make no theoretical assumptions about the behavior of the phenomenon under study. Thus, the relationship between system variables can be freely captured.

In this study, the models used included multiple linear regression models with stepwise forward selection of variables, multilayer perceptron neural networks and random forest regressors. Multiple linear regression (MLR) is a model based on minimizing the sum of squares of the deviations between observed and estimated values (least squares) and was used to help selecting variables for the artificial intelligence (AI) models and as a baseline in comparisons of results and performances with the AI methods, as suggested by [48]. For the MLR model, the feature selection process was performed stepwise forward using

the "Sequential Feature Selector" method [49] from the 'mlxtend' library [50] of Python programming language version 3.9.12 (https://pypi.org/project/mlxtend/). The feature selection ensures a better result for the MLR model, allowing greater robustness, using a smaller number of variables but with a greater relevance.

Regarding the artificial intelligence methods, it is worth referring to the performance of existing algorithms, which can be very different depending on the method's complexity and data quality [51]. In the present study, we selected multilayer perceptron neural networks and random forest regressor, using the same variables as those resulting from the stepwise selection of MLR models. Regardless of the model, the dissolved and total concentrations of aluminum, iron and manganese in water were used as targets (dependent variables; namely, Al, Fe and Mn from Table 1) and the remaining parameters included in the working dataset as features (independent variables; i.e., the other variables listed in Table 1). The multilayer perceptron (MLP) neural network model was chosen for this work because it allows training on nonlinear problems, although it has training difficulties due to local minimum problems and has a large number of hyperparameters, as mentioned in [52]. The Random Forest (RF) regressor model was chosen because it is very flexible in regression problems, for combining a collection of decision trees and being robust in relation to outliers. Its training is slower when compared with other machine learning algorithms, but the results are more precise and accurate [53]. The training method was identical for all models and cross-validation with 3 folds was used. The precision of all machine learning methods was assessed by the $R^2_{Adjust}$ (Equation (1)), as proposed by [54]:

$$R^2_{Adjust} = 1 - \frac{(1 - R^2) \times (n - 1)}{n - k - 1} \tag{1}$$

where $R^2$ is the coefficient of determination, n the number of entries in the dataset, and k the number of independent variables.

$R^2_{Adjust}$ is the percentage of the model's ability to correctly estimate the dependent variable as a function of the independent variables. The closer it is to 1, the better the model.

The accuracy was analyzed by the Root Mean Squared Error (RMSE) (Equation (2)) [55]:

$$RMSE = \sqrt{\frac{\sum_{i=1}^{N}(Yobs_i - Yest_i)^2}{N}} \tag{2}$$

where Yobs is the target's measured value and Yest the value estimated by the machine learning models. RMSE is the average error between the predicted and observed values in a data set. The smaller the value, the smaller the error.

Taking the statistical and machine learning results altogether, the river model is expected to provide an assortment of insights about water–sediment interactions valid for the working spatial and temporal domains, which comprise the contribution of sediment source composition and grain size to changes in river water composition and the influence of streamflow discharge and environmental conditions on the type of transport (dissolved particulate). In similar work, these insights, together with the neural networks trained by the machine learning algorithms, will set up the basis for predictions of water composition as function of variations in streamflow and sediment characteristics. Thus, the river model as whole (Figure 2) provides a comprehensive assessment and a panoramic view over most factors controlling water–sediment interactions in a large river. It goes without saying that the specific methods used to implement the river model can change without compromising its structure or concept.

## 3. Results and Discussion

*3.1. Experimental Results*

3.1.1. Descriptive Statistics

- Aluminum, iron and manganese concentrations in water

The mean concentrations of heavy metals in the dissolved phase (Al(dis), Fe(dis) and Mn(dis)) were always lower in the dry period than in the rainy period (Figure 3). This outcome points to a positive correlation of these concentrations with the flow rate (cf. Spearman correlation in Section 3.1.2), recalling that the discharge from Paraopeba River basin is much lower in the dry period than in the rainy period. The contrast between higher metal concentrations in the high-flow period (rainy) relative to those in the low-flow period (dry) has been noted in other regions worldwide, notably in north-central Arkansas (USA), as reported in the study of [56]. It implies the absence of dilution effects, notwithstanding the report of those effects in other studies [57,58].

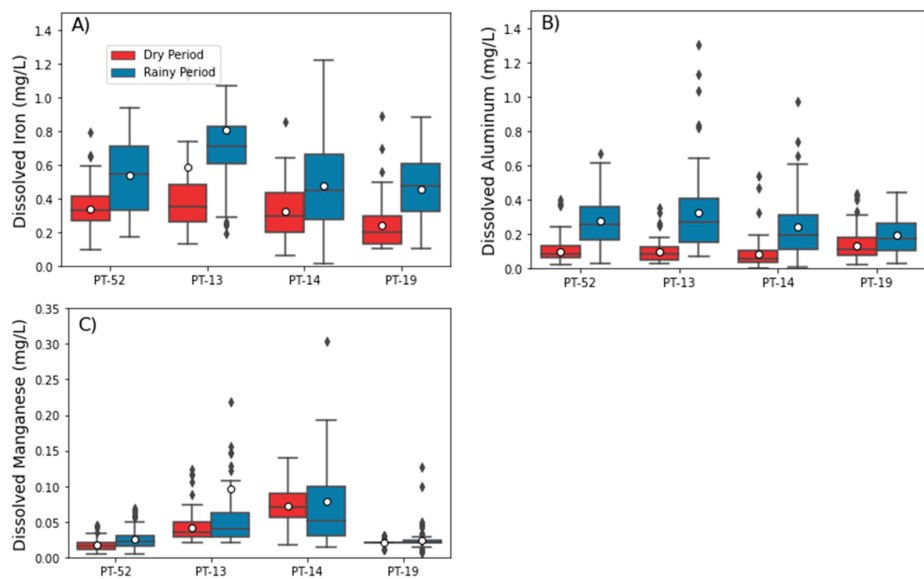

**Figure 3.** Boxplot diagrams of: (**A**) Fe(dis), (**B**) Al(dis) and (**C**) Mn(dis) concentrations in the dry and rainy periods of 2019 to 2021, measured at the "upstream" (PT-52), "anomalous" (PT-13 and PT-14) and "natural" (PT-19) monitoring stations.

The "anomalous" monitoring stations (PT-13 and PT-14) always presented higher concentrations of those dissolved metals (Al(dis), Fe(dis) and Mn(dis); Figure 3) than the "upstream" station. On the other hand, the station PT-13, which received the discharge from the B1 dam break, showed higher values of iron and aluminum compared with the station PT-14 and the "natural" station. The station PT-14 revealed higher manganese concentrations than the other stations, and the "natural" station always obtained the lowest concentrations of Al(dis), Fe(dis) and Mn(dis), regardless of whether the period was dry or rainy. Overall, these results suggest a spread of contamination downstream, without affecting the zone where the "natural" station is located, but increasing the manganese concentrations in the station PT-14, where the Igarapé weir is located. The dissolved fraction of metal concentrations at the station PT-13 unequivocally marks the disturbance caused by the mine tailings. The accompanying magnitude can be related to the increase of Fe(dis), Al(dis) and Mn(dis) concentrations, between the "upstream" station and "anomalous" station PT-13. Thus, in the rainy period, at the "upstream" station, the average value of Fe(dis) was 0.55 mg·L$^{-1}$, while at the station PT-13, it was 0.8 mg L$^{-1}$. Therefore, an average increase of 0.25 mg L$^{-1}$ [+45%] occurred due to the break of dam B1. Regarding the Al(dis), which showed values of 0.7 mg L$^{-1}$ and 0.55 mg L$^{-1}$, respectively, at the "anomalous" PT-13 and "upstream" stations, the increase was 0.15 mg L$^{-1}$ [+27%]. Finally,

for the Mn, where the concentrations varied between 0.1 mg L$^{-1}$ and 0.025 mg L$^{-1}$ from the "anomalous" PT-13 to the "upstream" station, the increase was 0.075 mg L$^{-1}$ [+300%]. In the dry period, the increases were smaller than in the rainy period, namely, 0.25 mg L$^{-1}$, 0 and 0.024 mg L$^{-1}$, for Fe(dis), Al(dis) and Mn(dis), respectively. The observed increases likely reflect the relative mobility of Al, Fe and Mn in the river water, which depends on the isolated or combined action of several factors (e.g., ionic radius) and processes (cation exchange, adsorption-desorption, hydrolysis, oxidation–reduction). Thus, the large difference between the percentages of manganese increases relative to the iron and aluminum counterparts probably reflect the greater mobility of Mn relative to Fe and Al, as recognized in the reference literature [59].

The concentrations of Al(dis), Fe(dis) and Mn(dis) were always lower than the respective total concentrations (Al(tot), Fe(tot) and Mn(tot)), as can be seen by comparing Figures 3 and 4. On the other hand, the differences between total and dissolved concentrations were more expressive in the rainy season relative to the dry period. Similar outcomes were obtained in the work of Teramoto [40], who performed in-situ DGT (diffusive gradient in thin films) monitoring and desorption experiments in the Paraopeba River after the collapse of the B1 dam, which corroborates our results. These authors also pointed a justification for the results. They anticipated that more material is moved in suspension during the rainy season, which can raise the total concentrations of all metals. In addition, a portion of the increased river flow during the rainy season is linked to surface runoff, which may reduce the amounts of dissolved and labile metals. Similarity among our results and those obtained elsewhere can also be used to validate our findings. Thus, higher percentages of particulate versus dissolved transport were also reported in studies in the Athabasca River [60], in the Fukushima region, where the accident at the Fukushima Daiichi Nuclear Power Station occurred [61], or along the Yangtze River basin from its source to the estuary during flood and drought periods [62], among others. In some case studies and for some metals (e.g., aluminum), the proportions of dissolved transport overlapped those of particulate transport, mainly when the effect of water pH was prominent over the other factors. Usually, those observations hold for rivers affected by acid mine drainage, where pH is low (<5), namely, for the Odiel River in southwestern Spain [63]. However, that was not the case in the present study.

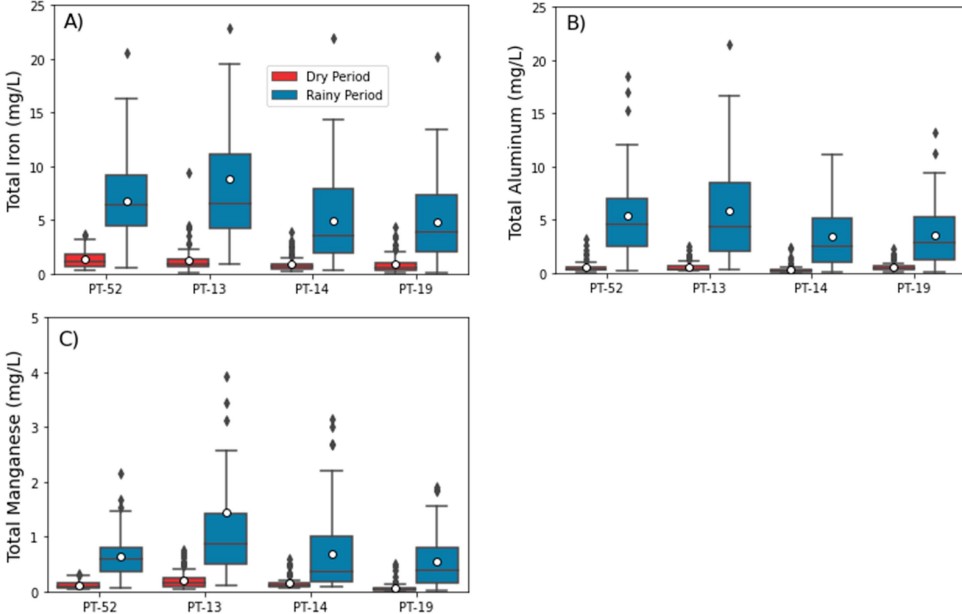

**Figure 4.** Boxplot diagrams of: (**A**) Fe(tot), (**B**) Al(tot) and (**C**) Mn(tot) concentrations in the dry and rainy periods of 2019 to 2021, measured at the "upstream" (PT-52), "anomalous" (PT-13 and PT-14) and "natural" (PT-19) monitoring stations.

The average concentrations of Al(tot), Fe(tot) and Mn(tot) were always lower in the dry period than in the rainy period (Figure 4). This outcome reveals a positive correlation of those concentrations with the flow rate (cf. Spearman correlation in Section 3.1.2), but higher than those previously detected for the dissolved concentrations. The effect of flow rate on increasing total concentrations, which can be related with particulate transport, was also observed in the study of Beltaos and Burrel [64] in the Saint John River (Canada), where slight increases in metal concentrations were reported during the thaw period. As verified with the dissolved fraction concentrations, the "anomalous station" monitoring station always had higher concentrations of Al(tot), Fe(tot) and Mn(tot) than the "upstream" station. Thus, one can also estimate how much the rupture of dam B1 impacted the total concentrations, by computing the differences of Al(tot), Fe(tot) and Mn(tot) between the two stations. In the rainy period, the average Fe(tot) at the "upstream" station was 7 mg $L^{-1}$, while at the "anomalous station", it was 9 mg $L^{-1}$. This represents an average increase of 2 mg $L^{-1}$ [+28%]. Regarding the Al(tot), where the values were 6 mg $L^{-1}$ and 5.5 mg $L^{-1}$, respectively, at the "anomalous" and "upstream" stations, the increase was 0.5 mg $L^{-1}$ [+9%]. And for the Mn(tot) (with 1.5 mg $L^{-1}$ and 0.7 mg $L^{-1}$), the increase was 0.8 mg $L^{-1}$ [+114%]. These results are similar to (though smaller than) those verified with the dissolved concentrations, so they are interpreted in a similar way. In the dry period, the increases were null, with the exception of Mn, which increased 0.1 mg $L^{-1}$. The "natural" station showed values of total concentrations not very different from those measured at the "upstream" station, indicating that the "natural" station has not yet undergone significant changes due to the dam break.

- Ambient conditions and concentrations of other elements in the water

Regarding the variables pH (Figure 5A), temperature (Figure 5B), turbidity (Figure 5C) and total arsenic (Figure 5D), they did not show significant differences between the "upstream" and "anomalous" monitoring stations. In other words, these variables were not typical markers of the B1 dam break. However, dissolved lead (Figure 5E), total lead (Figure 5F), dissolved phosphorus (Figure 5G) and total phosphorus (Figure 5H) showed differences between the "upstream" and "anomalous" stations. Thus, as observed with iron and manganese, lead and phosphorus also left a signature of Brumadinho's tailings dump in the Paraopeba River, at least between January 2019 and December 2021.

When comparing the results between the dry and rainy periods, there are significant contrasts between temperature (Figure 5B), turbidity (Figure 5C), dissolved phosphorus (Figure 5G) and total phosphorus (Figure 5H). The results for temperature and turbidity are not surprising since they likely reflect the differences in air temperature and flow, respectively, which are usually higher in the rainy period relative to the dry period of Brazil's tropical climate. The higher values of phosphorus in the rainy period, both dissolved and total, reinforce the belief that possible dilution effects related with streamflow rate increases were not dominant in this study. Thus, the justification for the results should be found in the resuspension of particulate phosphorus in response to higher flows observed in the rainy period, as well as in the subsequent transfer of this element to the dissolved phase (e.g., through desorption). Considering that phosphorus, in addition to its origin in the tailings mud, is mainly related to agricultural activities, the higher concentrations observed in the rainy period may also reflect a greater erosion of phosphorus from crop lands, transport in surface runoff, and discharge into the main watercourse (Paraopeba River).

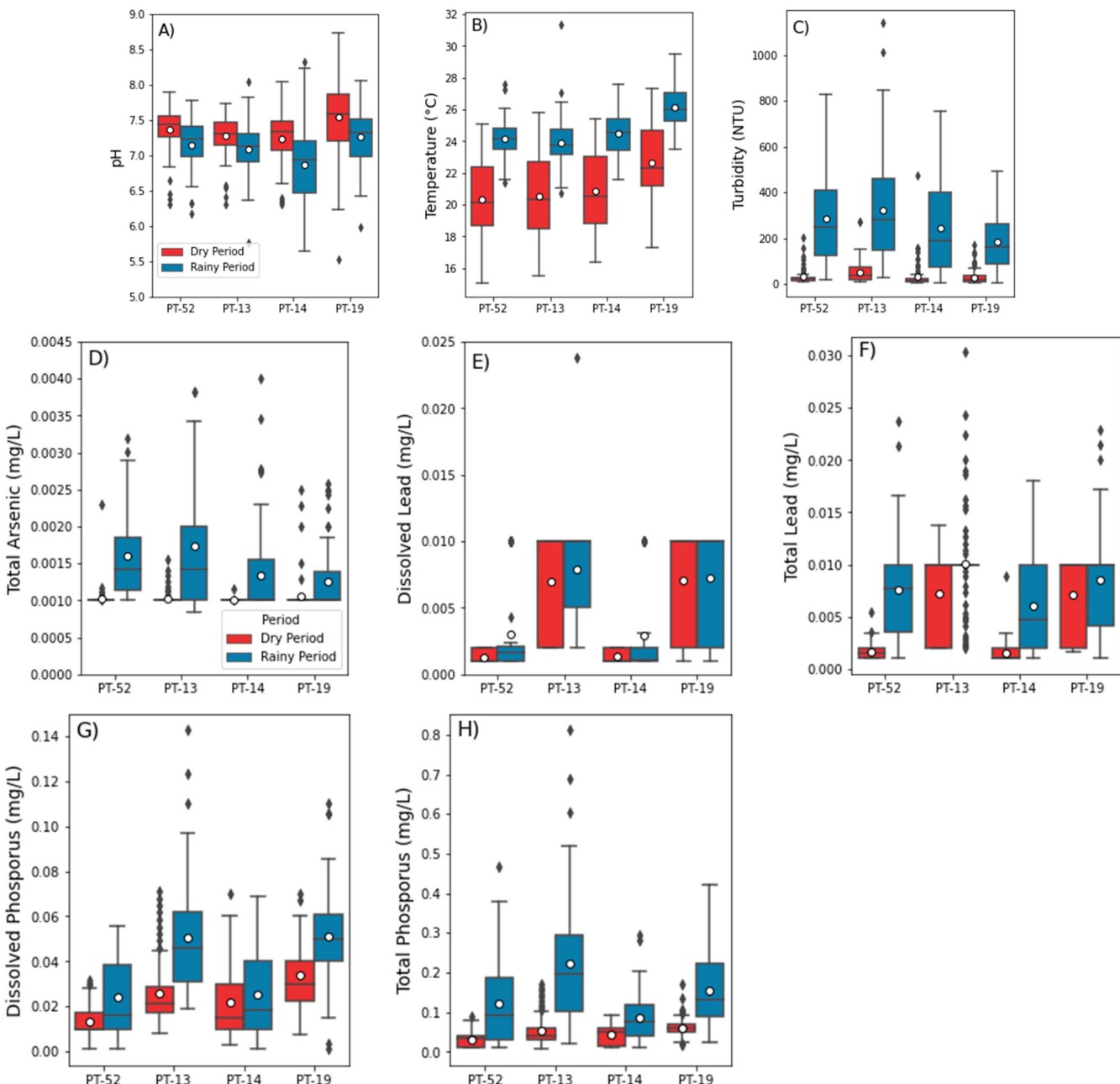

**Figure 5.** Boxplot diagrams of ambient conditions and contaminants other than Fe, Al and Mn dissolved and total concentrations in the water of Paraopeba River: (**A**) pH, (**B**) temperature, (**C**) turbidity, (**D**) total arsenic, (**E**) dissolved lead, (**F**) total lead, (**G**) dissolved phosphorus, (**H**) total phosphorus, in the dry and rainy periods of 2019 to 2021, measured at the "upstream" (PT-52), "anomalous" (PT-13 and PT-14) and "natural" (PT-19) monitoring stations.

- Metal, arsenic and phosphorus concentrations in sediment and tailings

In the (E) compartment of Table 1, related to the chemical composition of sediments + tailings, the "anomalous" PT-13 station showed the lowest concentration of aluminum (Figure 6A), while the concentrations of iron (Figure 6D), phosphorus (Figure 6E) and manganese (Figure 6F) increased from the "upstream" to the "anomalous" PT-13 station and from the latter to the "anomalous" PT-14 station, which reached the highest values in both the dry and rainy periods. These results confirm aluminum as a secondary contributor to the tailings, and that, during the January 2019–December 2021 period, a spreading of

iron- and manganese-rich tailings from the zone most impacted by the spill to the Igarapé weir zone located nearly 50 km downstream (Figure 1) has occurred.

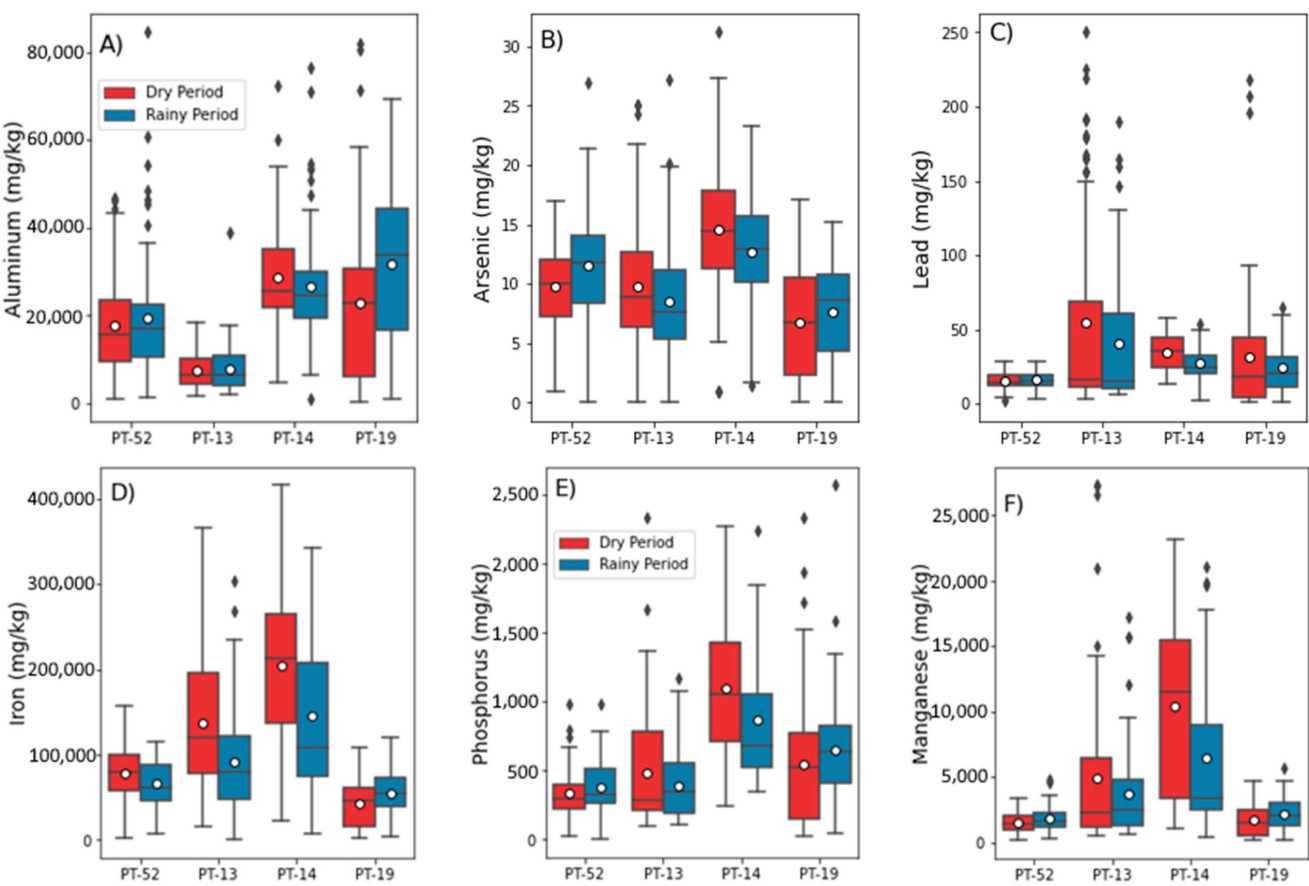

**Figure 6.** Boxplot diagrams of compartment (**E**) variables, which are related to the chemical composition of sediments + tailings mixtures: (**A**) aluminum, (**B**) arsenic, (**C**) lead, (**D**) iron, (**E**) phosphorus and (**F**) manganese, in the dry and rainy periods of 2019 to 2021, measured at the "upstream" (PT-52), "anomalous" (PT-13 and PT-14) and "natural" (PT-19) monitoring stations.

- Particle size distribution in sediments and tailings

The studied period showed an average flow rate of 90 m$^3$ s$^{-1}$ and 25 m$^3$ s$^{-1}$ in the rainy and dry periods, respectively (Figure 7H). This flow difference did not significantly alter the particle displacement between the two seasons, with the exception of the "upstream" station in relation to the very fine-grained (Figure 7C) and fine-grained (Figure 7D) sand fractions. The coarse-grained (Figure 7F) and very coarse-grained (Figure 7G) sands had higher concentrations in the dry period, mainly in the "anomalous" PT-13 and "natural" monitoring stations. In these cases, the asymmetry of dry period values was much larger than that of their rainy period counterparts. Eventually, in the rainy period, the higher flows ensure the displacement of all sediment fractions, reducing the asymmetry, while in the dry period, there is a selective deposition of coarse particles, justifying the larger asymmetry [65,66].

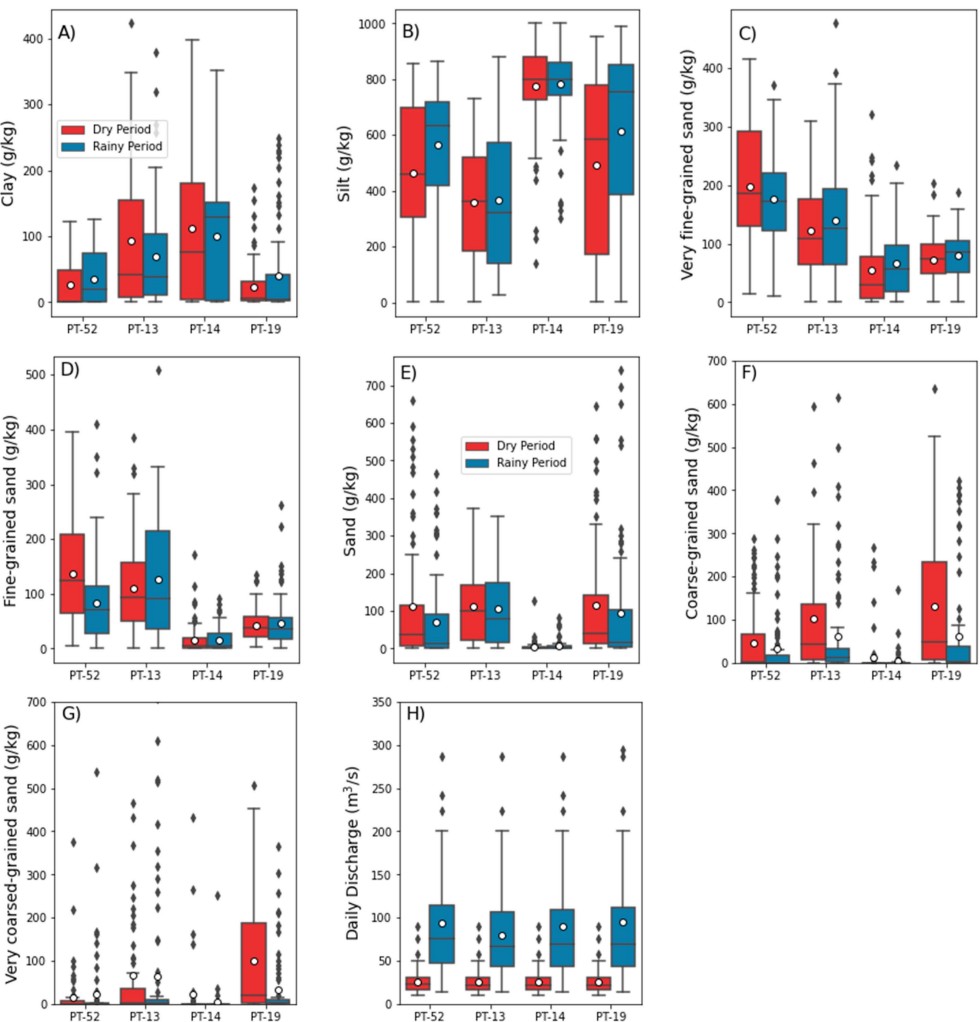

**Figure 7.** Boxplot diagrams of (F) compartment variables (Table 1), which are related to the granulometric fractions of sediments and tailings: (**A**) clay, (**B**) silt, (**C**) very fine-grained sand, (**D**) fine-grained sand, (**E**) sand, (**F**) coarse-grained sand, (**G**) very coarse-grained sand. Boxplot diagram of (**D**) compartment variable: (**H**) river flow. The dry and rainy periods between 2019 and 2021 were considered, as well as the "upstream" (PT-52), "anomalous" (PT-13 and PT-14) and "natural" (PT-19) monitoring stations.

### 3.1.2. Analytical Statistics

- Spearman's rank-order correlation matrix

The correlation analysis between all the variables listed in Table 1, computed for the "upstream" and "anomalous" stations and the dry and rainy periods (Figure 8), points to an overall negative correlation between the (E) compartment, which refers to the chemical characteristics of sediments, and the streamflow compartment (D). The sediment's granulometry compartment (F) also correlated with the flow, but the results depended on the season. In the dry period, the finer fractions correlated negatively and the coarser ones positively with the flow. In the rainy period, the correlations with the flow were generally negative, except for the clay fraction at the "upstream" station. Finally, the compartments concerning the concentrations in water, dissolved (A) or total (B), had generally positive correlations with the streamflow. The correlations between flow and grain size suggest that mobilization of particles from the river bed into the water column can be markedly influenced by particle diameter, as noted in other works [67–69], meaning that cohesive properties of finer particles [70] also present in the sediments and tailings may not play a significant role in that process in the studied locations. In a recent publication where sediment

+ tailings transport was modeled in a sector of the Paraopeba River using the physically based Riverflow 2D computer package [71], namely around the Igarapé weir, located near the PT-14 station, the results were reliable without considering the aforementioned cohesive effects. Thus, the finer particles were moved from the bed to the water column under the action of flow (decreasing their relative concentration in the mixture), while the coarser ones remained immobile (increasing their relative concentration). In the rainy period of the "upstream" station, the relative increase in the clay fraction points to cohesive effects. In this period, the transport capacity of higher flows likely allowed the mobilization of all grain size fractions simultaneously, facilitating the interaction between particles, namely, between clay particles that eventually developed cohesive bonds [72]. Thus, the non-clay fractions maintained a non-cohesive behavior, responding to flow increases the same way as in the dry period, while the clay fraction's comportment reversed due to cohesive forces. Regardless of the season, the resuspension of sediments was accompanied by the mobilization of metals. The correlation between flow and sediments (compartment (F)), coupled with the correlation between flow and metals and other chemical elements present in the sediments (E) or in the water (A and B), probably means that, after resuspension, the chemical elements of sediment particles moved to the water column in particulate or dissolved form, through processes of dissolution or desorption [40]. This transfer is further corroborated by the negative correlations observed between concentrations in water and concentrations in sediment, particularly evident at the "upstream" station (Figure 8A,B).

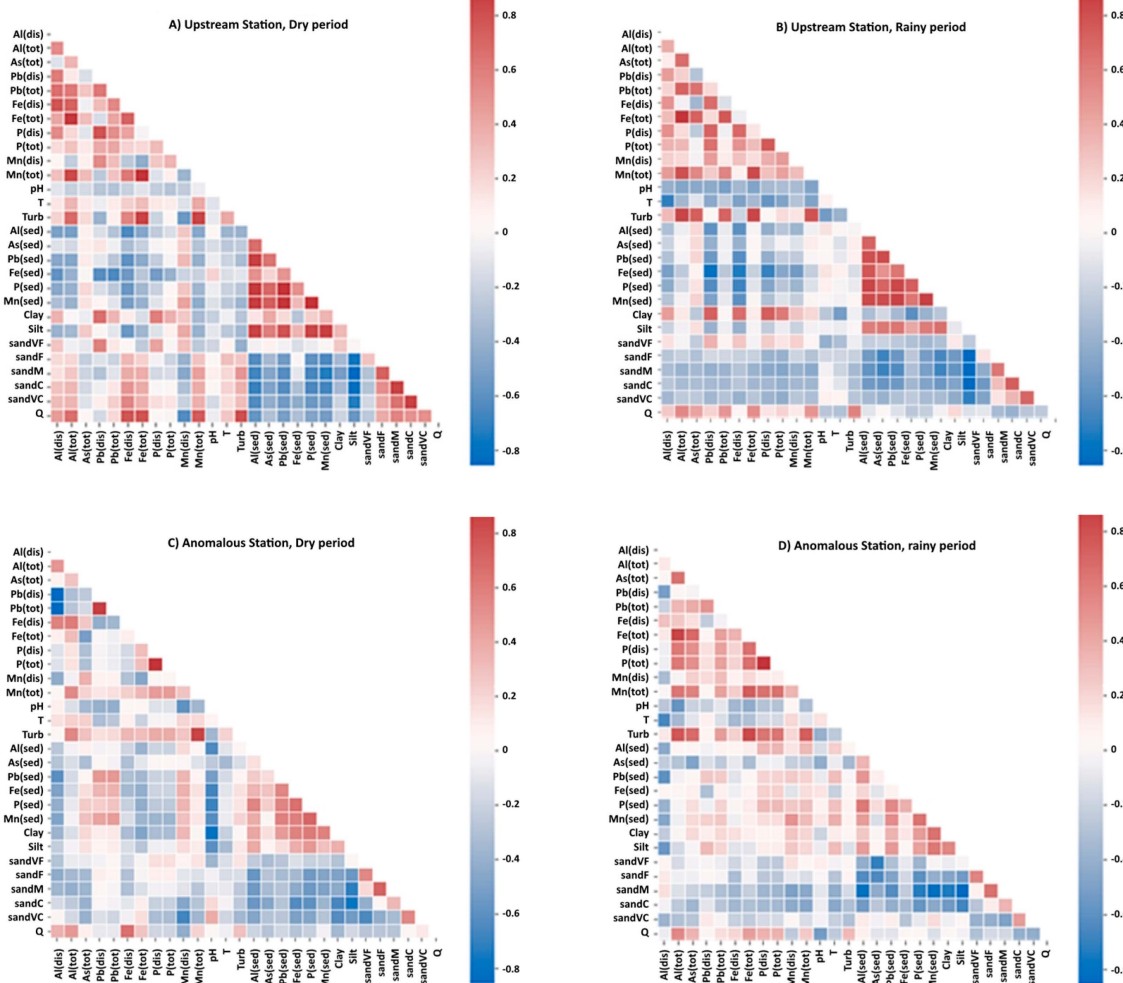

**Figure 8.** Spearman's rank-order correlations between all variables listed in Table 1, computed at the "upstream" (panels (**A**,**B**)) and "anomalous" PT-13 (**C**,**D**) monitoring stations, in the dry and rainy periods.

Considering the results of Spearman's rank-order correlation analysis (Figure 8), a selection of five variables with the highest positive and negative correlations for the prediction of Al, Fe and Mn concentrations in water, dissolved and total, was carried out for each monitoring station and the two seasons (Table 2). For example, total aluminum, total and dissolved iron and flow rate (in green) are the variables with the highest correlations for estimating the aforementioned targets at the "upstream" station and in the dry period. In general. the results highlight the inter-correlation among the three metals, probably of geological origin [39], as well as their preferential association with turbidity and temperature in the rainy period.

**Table 2.** List of the highest Spearman rank-order correlation coefficients at the "upstream" and "anomalous" PT-13 monitoring stations and in the dry and rainy periods.

| Station | Period | Target Variable | Greater +Correlation | Feature | Greater −Correlation | Feature | Most Important Features | |
|---|---|---|---|---|---|---|---|---|
| PT-52 | Dry SeaSon | Al(dis) | 0.725 | Fe(dis) | −0.490 | Fe(sed) | 0.913 | Turb |
| | | Al(tot) | 0.913 | Turb | −0.315 | Al(sed) | 0.896 | Al(tot) |
| | | Fe(dis) | 0.745 | Q | −0.639 | Al(sed) | 0.895 | Fe(tot) |
| | | Fe(tot) | 0.896 | Al(tot) | −0.414 | Al(sed) | 0.745 | Q |
| | | Mn(dis) | 0.426 | Silt | −0.416 | Q | 0.725 | Fe(dis) |
| | | Mn(tot) | 0.895 | Fe(tot) | −0.402 | Al(sed) | | |
| PT-52 | Rainy Season | Al(dis) | 0.665 | P(dis) | −0.479 | T | 0.967 | Fe(tot) |
| | | Al(tot) | 0.967 | Fe(tot) | −0.234 | pH | 0.967 | Al(tot) |
| | | Fe(dis) | 0.817 | P(dis) | −0.578 | Fe(sed) | 0.817 | P(dis) |
| | | Fe(tot) | 0.967 | Al(tot | −0.238 | pH | −0.578 | Fe(sed) |
| | | Mn(dis) | 0.469 | P(tot) | −0.250 | Sand_c | −0.479 | T |
| | | Mn(tot) | 0.838 | Fe(tot) | −0.244 | T | | |
| PT-13 | Dry Season | Al(dis) | 0.729 | Al(tot) | −0.466 | Pb(dis) | 0.898 | As(tot) |
| | | Al(tot) | 0.729 | Al(dis) | −0.326 | pH | 0.831 | Turb |
| | | Fe(dis) | 0.898 | As(tot) | −0.185 | Sand_f | 0.729 | Al(tot) |
| | | Fe(tot) | 0.557 | Mn(tot) | −0.353 | pH | 0.729 | Al(dis) |
| | | Mn(dis) | 0.686 | As(tot) | −0.281 | Sand_vc | 0.557 | Mn(tot) |
| | | Mn(tot) | 0.831 | Turb | −0.285 | pH | | |
| PT-13 | Rainy Season | Al(dis) | 0.766 | Fe(dis) | −0.387 | T | 0.908 | Turb |
| | | Al(tot) | 0.743 | Turb | −0.289 | pH | 0.886 | Fe(tot) |
| | | Fe(dis) | 0.885 | Mn(dis) | −0.247 | As(sed) | 0.885 | Mn(dis) |
| | | Fe(tot) | 0.908 | Turb | −0.205 | T | 0.885 | Fe(dis) |
| | | Mn(dis) | 0.885 | Fe(dis) | −0.259 | As(sed) | −0.387 | T |
| | | Mn(tot) | 0.886 | Fe(tot) | −0.228 | As(sed) | | |

- Principal Component Analysis

The results of principal component analysis (PCA) provided joint information (multivariate) about the list of parameters represented in Table 1, allowing a deeper understanding on the interactions among the defined compartments. The PCA was run four times, considering the combinations of monitoring stations and seasons, namely the "upstream" and "anomalous" stations and the dry and rainy periods. The dataset variability at the "upstream" station in the dry period, was explained in 50% by the components PC1 and PC2. Ata the same station, but in the rainy period, the percentage was 48.3%, while the "anomalous station" reached values of 40.6% and 46.3% for the dry and rainy periods, respectively (Figure 9).

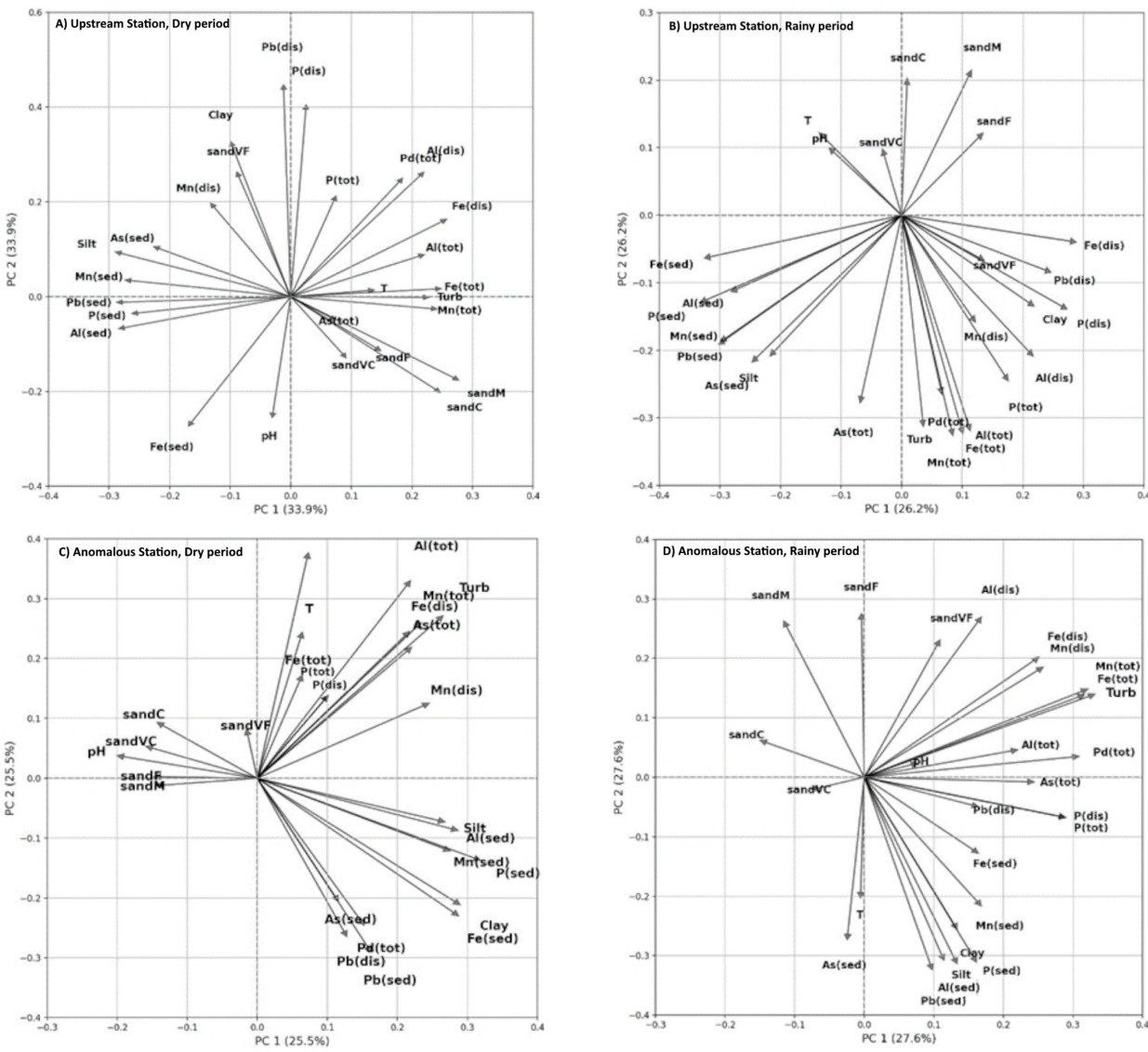

**Figure 9.** Principal component analysis (biplots) of all parameters listed in Table 1: (**A**) "upstream" station in the dry period; (**B**) "upstream" station in the rainy period; (**C**) "anomalous" PT-13 station in the dry period; (**D**) "anomalous" PT-13 station in the rainy period.

For the "upstream" station in the dry period (Figure 9A), the results suggested that the principal component PC1 should be termed "Chemical Characteristics", as the chemistry of the sediments (compartment (E); Table 1) clustered together, contrasting with the total concentrations of metals in water (compartment (B)). In addition, the principal component PC2 could be called "Granulometric Characteristics", because the concentrations of metals in the dissolved phase (compartment (A)) were in the opposition to the grain size of sediments (Compartment (F)). These results indicate for the "upstream" station in the dry period, that the Al(tot), Fe(tot) and Mn(tot) correlated negatively with the sediments' chemical characteristics, while the Al(dis), Fe(dis) and Mn(dis) had negative correlation with the granulometric characteristics. In terms of interactions among compartments, the results showed that the total concentrations in water are mostly explained by the resuspension of particles composed of aluminum, iron and manganese oxides (under the action of flow), while the dissolved concentrations are explained by the desorption (controlled by oxidation–reduction processes) of metals from sediment particles suspended in the water column that do not necessarily have to be aluminum, iron or manganese oxides. This type of nexus, involving total concentrations, dissolved concentrations and

their respective control factors (resuspension, change in the ambient conditions) has been similarly described in several rivers [73–75]. The results of PCA added comprehension to the boxplot and Spearman's correlation analyses, where this type of association has been suggested as well, because the PCA set a clear separation between particulate and dissolved transport through linking the two processes to independent variation trends (the PCs), while the previous analyses could not accomplish that result.

At the "upstream" station, but in the rainy period (Figure 9B), the principal components PC1 and PC2 could also be called "Chemical Characteristics" and "Granulometric Characteristics", respectively, similarly to the dry period. However, in the rainy period, the Al(dis), Fe(dis) and Mn(dis) in water showed negative correlation with the sediment's chemistry, while the Al(tot), Fe(tot) and Mn(tot) correlated negatively with the sediment's grain size. Thus, the associations among compartments reversed in the rainy period relative to the dry period. The main difference between the two periods is represented by the hydrologic regime, which is generically laminar and of low flow in the first case and turbulent and of high flow in the second. Thus, the observed inversion should be linked to that condition. The negative correlation between the chemical composition of water (dissolved metal concentrations) and the sediment's chemistry suggests that for the "upstream" station in the rainy period, resuspension succeeded by dissolution of Al, Fe and Mn oxides may have been a dominant mechanism in the partition and transport of these metals. As stated, during periods of high stream discharge, the flow may become turbulent, a condition that causes the disaggregation of large cohesive flocs through more intense and frequent collisions, thus increasing the dissolution potential of Al, Fe and Mn oxides [76,77]. On the other hand, the higher flows from the rainy period allow the re-suspension of coarser particles that are less capable of metal adsorption because of their smaller specific surface area [78,79]. This could help to explain the negative correlations between the sediment's granulometric compartment and the total concentrations of metals in the water.

The "anomalous" station obtained very different responses from those observed at the "upstream" station representing the non-impacted (control) sector of Paraopeba River. Thus, the differences between the two stations should be viewed as the impact of the B1 dam rupture on the structure of the relationships between the flow (either in the dry or rainy period), the characteristics of sediments + tailings mixtures, and the quality of river water. In the dry period, PC1 opposed the grain size classes to all the other variables in the system. In PC2, there was a negative correlation between the dissolved and total concentrations of metals in the water and the concentrations of metals in the sediments + tailings (Figure 9C). In the rainy period, there was a strong negative association in PC1 with the percentage of sand and the other system variables. PC2 opposed the concentration of lead in the sediments + tailings to the concentrations of the other variables (Figure 9D). Regardless of the period being dry or rainy, PC1 highlights the importance of grain size for the structure of "anomalous station" data. This is not surprising because this "anomalous" station received sediments and tailings during the studied period, and a major difference between natural sediments and tailings is represented by their grain size distributions. The tailings that were injected into the Paraopeba River were characterized by a mixture of silt (63.4%) and clay (36.3%), while the natural sediments contained some silt (46.9–58.3%; dry period-rainy period) but were very poor in clay (2.3–3.9%). Regarding PC2, in the dry period, the results point out the dominance of sediments + tailings chemistry in the formation of contaminant concentrations in water, both of particulate and dissolved fractions. Combining these results with those of PC1 for this period, it becomes clear that, even under conditions of low flow, in the impacted zone, it was possible to mobilize the tailings from the bed into the water column, increasing the total concentrations (particulate fraction) downstream, and simultaneously desorb or dissolve the metals contained in the silt and clay particles, increasing the concentrations of the dissolved fraction. In other words, the dry period results expose under no doubt an impact of the B1 dam rupture on the water quality of the Paraopeba River, through the resuspension of iron- and manganese-rich silts and clays from the tailings and subsequent mobilization of these metals into the water column.

### 3.2. *Artificial Intelligence Modelling*

Artificial Intelligence Methods

The experimental phase of this study statistically analyzed the processes and conditions responsible for the concentrations of aluminum, iron and manganese in the impacted and non-impacted areas of Paraopeba River after the occurrence of B1 dam disaster at the Córrego do Feijão mine, which injected 2.8 $Mm^3$ of iron- and manganese-rich tailings into the main watercourse on 25 January 2019. One way to understand how these concentrations may evolve in the future is to develop models capable of being precise and accurate enough to gain predictive power. The following sections present the results of the models tested with increasing complexity, which have always shown remarkable precision and accuracy. They will therefore serve as a basis for the development of scenarios to support decision-making in the river's ecological and environmental restoration.

- Multiple linear regression with stepwise forward selection of variables

The multiple linear regression (MLR) with stepwise forward selection of variables was precise for the estimates of dissolved and total Al, Fe, Mn, because the average $R^2_{Adjust}$ was 0.84, with all models significant at $p < 0.05$. The models behaved better for iron ($R^2_{Adjust}$ = 0.89), than for aluminum (0.84) and finally for manganese (0.77). These numbers jointly refer to precisions of total and dissolved concentrations taken together. In more detail, it was seen that the models behaved better for the total concentrations relative to the dissolved concentrations, because the corresponding average $R^2_{Adjust}$ were larger in the first case, namely for Al (0.92 > 0.75), Fe (0.96 > 0.81) and Mn (0.90 > 0.63). The accuracy values were all less than 2 for Al(tot), Fe(tot) and Mn(tot) and less than 0.2 for Al(dis), Fe(dis) and Mn(dis). The $R^2_{Adjust}$ of all MRL models are depicted in Table A1 of Appendix A.

The feature variables selected for the MLR models differed according to the target variable, namely for the total and dissolved concentrations of aluminum, iron and manganese. They are indicated in Table A1 of Appendix A and accompanied by the corresponding fitting parameters. For example, to estimate the dissolved Al at the "upstream" station, considering the joint dry and rainy periods, the fitted equation was (Equation (3)):

$$Al(dis) = 0.186 + 0.15 \, Al(tot) - 0.032 \, Pb(dis) + 0.029 \, Pb(tot) + 0.071 \, Fe(dis) - 0.127 \, Fe(tot) + 0.031 \, Pb(dis) + 0.008 \, Mn(dis) + 0.025 \, Turb + 0.001 \, sandM - 0.017 \, sandVC \quad (3)$$

As the variables were standardized, the angular coefficients also informed about their importance in the estimate. In the case of Al(dis) equated above, the most important variable was Al(tot), with an angular coefficient of 0.15. Thus, for each standardized increase of one unit in Al(tot), the Al(dis) will increase by 0.15 or 15%. The linear coefficient or intercept (0.186) indicates the mean value of Al(dis) at the monitoring station. The models for dissolved and total iron and for dissolved and total manganese are presented in Tables A2 and A3 in Appendix A, respectively.

Multiple linear regression gives good results, but it also has a high dependence on the variables. It is not always guaranteed that a strategy can estimate a variable before using it in a prediction equation.

- Artificial neural network multilayer perceptron

Anticipating potential non-linear relationships between variables from a riverine dataset (e.g., Table 1), we chose to apply machine learning models such as multilayer perceptron neural networks (MLP), with the purpose of improving the precision and accuracy attained by the multiple linear regression (MLR) models. The fit of MLP neural networks resulted in $R^2_{Adjust}$ = 0.79 (on average), which is similar than that of MLR models (0.84). However, the MLP fittings of "upstream" station applied to the rainy period were better (average $R^2_{Adjust}$ = 0.78) than the corresponding fittings based on MLR (0.74). As with the MLR, the average $R^2_{Adjust}$ decreased from the iron (0.84) to the aluminum (0.79) and then to the manganese (0.74) adjustments. The MLP model fits were implemented using the Grid Search CV method and resulted in different activation, architecture, learning

rate and solver models for each target (Table 3). The RMSE values (Equation (2)) for total and dissolved Al, Fe and Mn were less than 2 and 0.2, respectively.

**Table 3.** Multilayer perceptron neural networks for the estimation of dissolved and total aluminum (Al), iron (Fe) and manganese (Mn) concentrations in Paraopeba River water, assessed at the "upstream" and "anomalous" PT-13 stations and in the dry and rainy periods. The precision was computed by the $R^2_{Adjust}$ (Equation (1)). The accuracy values calculated by the RMSE (Equation (2)) were less than 2 for the total and less than 0.2 for the dissolved concentrations.

| Station/Period | Target | Activation | Architecture | Learning Rate | Solver | $R^2_{Adjust}$ |
|---|---|---|---|---|---|---|
| "Upstream" | Al(dis) | identity | 2, 2, 2 | adaptive | lbfgs | 0.79 |
| "Anomalous" PT-13 | Al(dis) | identity | 1, 3 | adaptive | lbfgs | 0.55 |
| "Upstream"/Dry | Al(dis) | identity | 2, 3, 4 | adaptive | lbfgs | 0.95 |
| "Anomalous" PT-13/Dry | Al(dis) | identity | 2, 1, 4 | adaptive | lbfgs | 0.85 |
| "Upstream"/Rainy | Al(dis) | identity | 1, 2, 2 | adaptive | sgd | 0.5 |
| "Anomalous" PT-13/Rainy | Al(dis) | identity | 1, 4, 1 | adaptive | sgd | 0.45 |
| "Upstream" | Al(tot) | identity | 3, 2 | adaptive | lbfgs | 0.98 |
| "Anomalous" PT-13 | Al(tot) | identity | 1, 2 | adaptive | sgd | 0.60 |
| "Upstream"/Dry | Al(tot) | identity | 2, 1 | adaptive | lbfgs | 0.99 |
| "Anomalous" PT-13/Dry | Al(tot) | identity | 2, 1 | adaptive | lbfgs | 0.97 |
| "Upstream"/Rainy | Al(tot) | identity | 2, 1 | adaptive | sgd | 0.96 |
| "Anomalous" PT-13/Rainy | Al(tot) | identity | 2, 1 | adaptive | lbfgs | 0.87 |
| "Upstream" | Fe(dis) | identity | 4, 3 | adaptive | lbfgs | 0.80 |
| "Anomalous" PT-13 | Fe(dis) | identity | 1, 2 | adaptive | sgd | 0.50 |
| "Upstream"/Dry | Fe(dis) | identity | 2, 1 | adaptive | lbfgs | 0.93 |
| "Anomalous" PT-13/Dry | Fe(dis) | identity | 2, 1 | adaptive | lbfgs | 0.95 |
| "Upstream"/Rainy | Fe(dis) | identity | 2, 1 | adaptive | lbfgs | 0.82 |
| "Anomalous" PT-13/Rainy | Fe(dis) | identity | 2, 1 | adaptive | sgd | 0.75 |
| "Upstream" | Fe(tot) | identity | 2, 4 | adaptive | sgd | 0.95 |
| "Anomalous" PT-13 | Fe(tot) | identity | 2, 2 | adaptive | sgd | 0.94 |
| "Upstream"/Dry | Fe(tot) | identity | 2, 1 | adaptive | lbfgs | 0.97 |
| "Anomalous" PT-13/Dry | Fe(tot) | identity | 2, 1 | adaptive | sgd | 0.52 |
| "Upstream"/Rainy | Fe(tot) | identity | 2, 2 | adaptive | sgd | 0.93 |
| "Anomalous" PT-13/Rainy | Fe(tot) | identity | 1, 2 | adaptive | lbfgs | 0.98 |
| "Upstream" | Mn(dis) | tanh | 4, 3 | adaptive | lbfgs | 0.52 |
| "Anomalous" PT-13 | Mn(dis) | tanh | 2, 2 | adaptive | lbfgs | 0.95 |
| "Upstream"/Dry | Mn(dis) | tanh | 2, 2 | adaptive | lbfgs | 0.45 |
| "Anomalous" PT-13/Dry | Mn(dis) | tanh | 2, 2 | adaptive | lbfgs | 0.31 |
| "Upstream"/Rainy | Mn(dis) | tanh | 2, 2 | adaptive | lbfgs | 0.54 |
| "Anomalous" PT-13/Rainy | Mn(dis) | tanh | 1, 2 | adaptive | lbfgs | 0.95 |
| "Upstream" | Mn(tot) | identity | 3, 2 | adaptive | lbfgs | 0.90 |
| "Anomalous" PT-13 | Mn(tot) | identity | 2, 1 | adaptive | lbfgs | 0.96 |
| "Upstream"/Dry | Mn(tot) | identity | 2, 2 | adaptive | lbfgs | 0.93 |
| "Anomalous" PT-13/Dry | Mn(tot) | identity | 2, 1 | adaptive | lbfgs | 0.87 |
| "Upstream"/Rainy | Mn(tot) | identity | 1, 2 | adaptive | lbfgs | 0.51 |
| "Anomalous" PT-13/Rainy | Mn(tot) | identity | 2, 2 | adaptive | sgd | 0.96 |

- Random forest

The results of MLR and MLP were precise and accurate, but showed some variability: the $R^2_{Adjust}$ differed among the metals and the RMSE differed among the dissolved and total concentrations. Thus, we implemented random forest (RF) regressor models based on decision trees (Table 4) with the purpose of overcoming these problems. The random forest regressor models obtained average $R^2_{Adjust} = 0.92$ and RMSE always lower than 0.2, regardless the target. These results indicate that the RF models were more accurate and precise than the MLR and MLP models.

**Table 4.** Random forest regressor model for the estimation of dissolved and total aluminum (Al), iron (Fe) and manganese (Mn) concentrations in Paraopeba River water, assessed at the "upstream" and "anomalous" PT-13 stations and in the dry and rainy periods. The precision was computed by the $R^2_{Adjust}$ (Equation (1)) and the average value was 0.92 regardless the target. The accuracy values calculated by the RMSE (Equation (2)) were always less than 0.2.

| Station/Period | Target | $R^2_{Adjust}$ |
|---|---|---|
| "Upstream" | Al(dis) | 0.94 |
| "anomalous" PT-13 | Al(dis) | 0.94 |
| "Upstream"/Dry | Al(dis) | 0.92 |
| "Upstream"/Rainy | Al(dis) | 0.92 |
| "anomalous" PT-13/Dry | Al(dis) | 0.88 |
| "anomalous" PT-13/Rainy | Al(dis) | 0.91 |
| "Upstream" | Al(tot) | 0.99 |
| "anomalous" PT-13 | Al(tot) | 0.94 |
| "Upstream"/Dry | Al(tot) | 0.92 |
| "Upstream"/Rainy | Al(tot) | 0.92 |
| "anomalous" PT-13/Dry | Al(tot) | 0.88 |
| "anomalous" PT-13/Rainy | Al(tot) | 0.91 |
| "Upstream" | Fe(dis) | 0.95 |
| "anomalous" PT-13 | Fe(dis) | 0.94 |
| "Upstream"/Dry | Fe(dis) | 0.92 |
| "Upstream"/Rainy | Fe(dis) | 0.92 |
| "anomalous" PT-13/Dry | Fe(dis) | 0.88 |
| "anomalous" PT-13/Rainy | Fe(dis) | 0.91 |
| "Upstream" | Fe(tot) | 0.98 |
| "anomalous" PT-13 | Fe(tot) | 0.94 |
| "Upstream"/Dry | Fe(tot) | 0.92 |
| "Upstream"/Rainy | Fe(tot) | 0.92 |
| "anomalous" PT-13/Dry | Fe(tot) | 0.88 |
| "anomalous" PT-13/Rainy | Fe(tot) | 0.91 |
| "Upstream" | Mn(dis) | 0.87 |
| "anomalous" PT-13 | Mn(dis) | 0.94 |
| "Upstream"/Dry | Mn(dis) | 0.92 |
| "Upstream"/Rainy | Mn(dis) | 0.92 |
| "anomalous" PT-13/Dry | Mn(dis) | 0.88 |
| "anomalous" PT-13/Rainy | Mn(dis) | 0.91 |
| "Upstream" | Mn(tot) | 0.97 |
| "anomalous" PT-13 | Mn(tot) | 0.94 |
| "Upstream"/Dry | Mn(tot) | 0.92 |
| "Upstream"/Rainy | Mn(tot) | 0.92 |
| "anomalous" PT-13/Dry | Mn(tot) | 0.88 |
| "anomalous" PT-13/Rainy | Mn(tot) | 0.91 |

*3.3. Summary*

Figure 10 summarizes the performance accomplished by the three machine learning models implemented in the "upstream" and "anomalous" PT-13 monitoring stations installed in the Paraopeba River (location indicated in Figure 1), in the dry and rainy periods. The Y-axis values refer to the average $R^2_{Adjust}$ of fittings to the Al(dis), Fe(dis), Mn(dis), Al(tot), Fe(tot) and Mn(tot) concentrations. The random forest regressor model performed better and with less variability than the other models. In absolute terms, the scores of random forest $R^2_{Adjust}$ were always higher than 0.9, ensuring a robust predictive capacity, even more so considering the accuracy values that were always lower than 0.2. To illustrate graphically this remarkable performance, Figure 11A–D plot the time series of measured and estimated Fe(dis) and Fe(tot) as assessed in the "upstream" and "anomalous" PT-13 stations, which show a considerable overlapping.

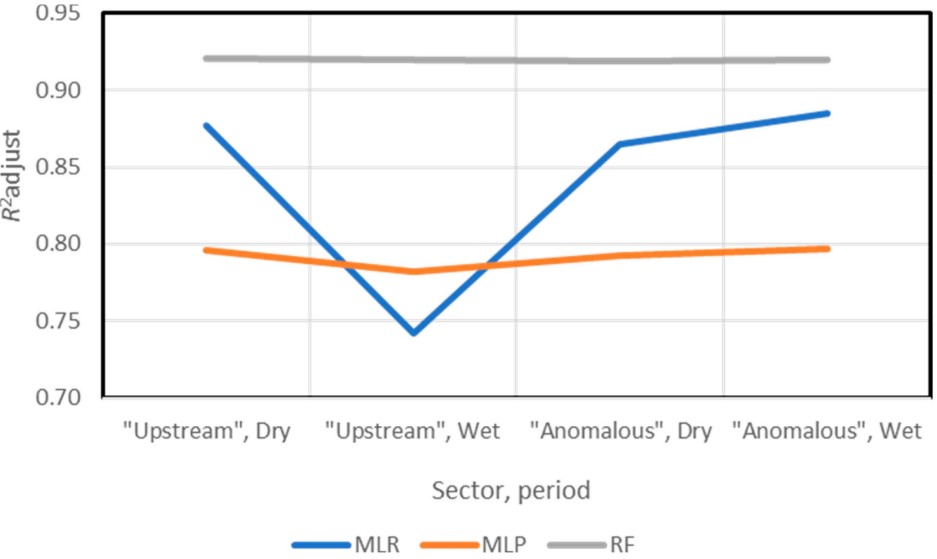

**Figure 10.** Mean values of $R^2_{Adjust}$ (Equation (1)) relative to the estimates of Al(dis), Fe(dis), Mn(dis), Al(tot), Fe(tot) and Mn(tot) concentrations, at the "upstream" and "anomalous" PT-13 stations and in dry and rainy periods. The terms MLR, MLP and RF designate the multiple linear regression models with stepwise forward selection of variables, multilayer perceptron neural networks and random forest regressor, respectively.

The Paraopeba River is an open and dynamic natural system that has suffered a large technological accident, making it a special case study. In fact, there have been more tailings dams failures in the world [80–83], but nothing that resembles the complexity of Paraopeba River's case, which, besides the accident, has, for example, numerous mining operations along its course. This characteristic means, among other things, that different dynamics occur along the main watercourse of this river system, which change the importance of variables in the mobilization of water and sediments, as well as their amplitudes in the dry and rainy periods, which are very contrasting.

In this study, we highlighted the performance of classical statistical analyses compared to machine learning-based analyses. The latter approach highlighted how complex the methods can be, but also warned of limitations and acknowledged very accurate results. The two approaches are complementary: the statistical assessment allows understanding the complexity of what a river system is—a complexity that is confirmed throughout the work by the numerous citations, both to similar studies reported worldwide as well as to studies focused on the Brumadinho disaster carried out by ourselves and other working groups.

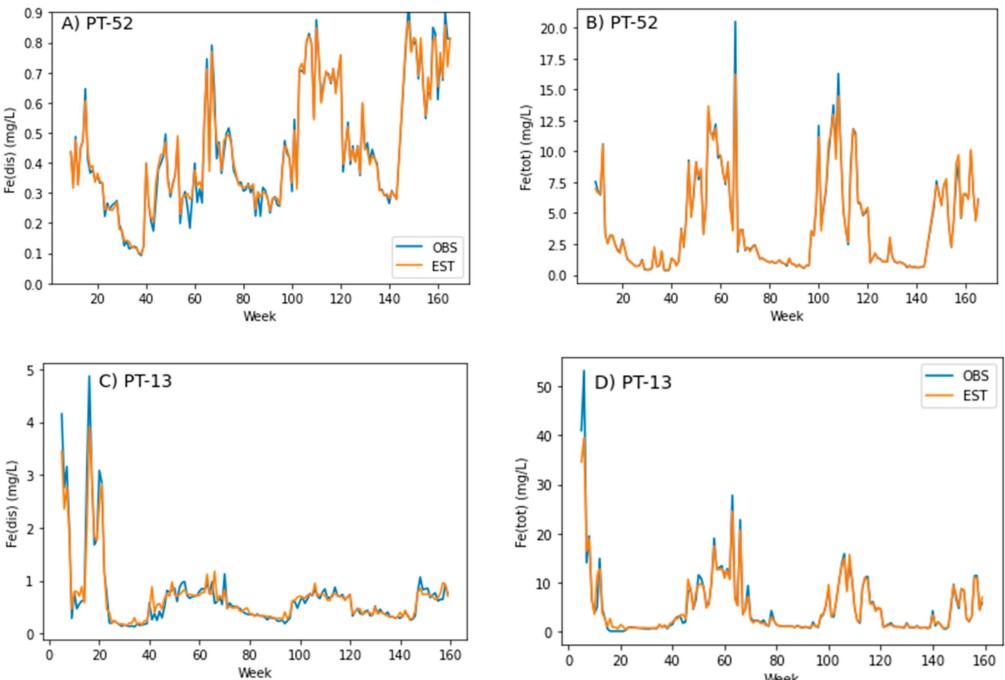

**Figure 11.** Time-series of Fe(dis) and Fe(tot) concentrations, as measured (obs in the legend) or estimated (est in the legend) by the random forest regressor model in the "upstream" (panels (**A**,**B**)) and "anomalous" PT-13 (panels (**C**,**D**)) stations.

### 3.4. Limitations, Implications and Future Work

The methodology presented has three main limitations: (1) it applies to rivers that have suffered a technological accident, resulting in the division of the river into three distinct segments. We have a non-impacted segment, an impacted segment and, finally, a potentially impacted segment. (2) The wastes are metals with water–sediment interactions. (3) The climate is tropical with a strong contrast in rainfall between the rainy and dry periods.

This article is part of a project called Entire which aims to study the impact of the tailings from the Brumadinho dam collapse for a restored aquatic environment and aims to model the three segments of the river to later be able to simulate the behavior of the river as a whole. The collapse of the Brumadinho dam has prompted a wide range of studies, which have resulted in numerous publications with valuable contributions in various areas, such as the economic–financial [84], health [85], social [86], environmental [86,87] and aquatic ecosystems [88].

In future work, the random forest regressor model will be used to predict the evolution of Al, Fe and Mn concentrations along the Paraopeba River, attempting to predict the time required for the "anomalous" sector to reach an environmental condition identical to the "upstream" sector (pre-rupture condition), and to see if (and how and when) it propagates to the "natural" sector, producing contamination.

### 4. Conclusions

The framework proposed in this study proved efficient to understand the spatial distribution of aluminum, iron and manganese concentrations in the Paraopeba River, currently segmented into sectors affected and not affected by the rupture of the B1 tailings dam, which exhibit marked differences in the grain size and composition of sediments. It also highlighted the role of streamflow, which is markedly different in the rainy period compared with the dry period. The sequence of results obtained by the univariate (boxplot diagrams), bivariate (Spearman rank-order correlation coefficients) and multivariate (principal component analysis) statistical analyses allowed an accumulation and integration of interpretations about the partition and transport of metals and other contaminants in the Paraopeba River, after the collapse of the mine-tailings dam B1 in Brumadinho in January 2019. On the other hand,

the comparative analysis of the precision and accuracy of three machine learning methods allowed us to identify the random forest regressor method as ideal for the elaboration of scenarios. Regarding the statistical analyses, it was concluded that:

1. The results of boxplots showed unequivocally that the concentrations of Al, Fe and Mn in the river water, either dissolved or total, were always lower in the dry period or at the "upstream" station, relative to the corresponding values in the rainy period or at the "anomalous" station. In addition, the dissolved fraction was systematically much lower than the particulate fraction. These results allowed us to highlight the effect of flow (intrinsically related to the season) on the resuspension and transport of sediments and tailings and the consequent spread of water contamination downstream, especially in the particulate form. They also made it possible to quantify the impact of the B1 dam breach on the increase of these metal concentrations: during the rainy period, the dissolved concentrations increased by 27–300% from the "upstream" to the "anomalous" station, and the total concentrations by 9–114%; during the dry period, the increases were smaller.

2. The Spearman rank-order correlation coefficients explained the effect of flow rate highlighted in the boxplots, as they generally allowed the establishment of a link between the increase in flow rate, the resuspension of sediment + tailings particles rich in metals, and the subsequent transfer of metals from the solid phase to the aqueous phase by dissolution or desorption. In addition, the correlation analysis allowed the identification of some peculiarities of sediment transport, namely, non-cohesive transport of clays during the dry period and cohesive transport of clays during the rainy period. Finally, it reinforced the conviction brought from the boxplot analysis that particulate transport dominates over dissolved transport because turbidity had one of the highest correlation coefficients with the flow and total metal concentrations.

3. The results of principal component analysis showed that the resuspension processes of natural sediments with adsorbed metals, as well as of iron and manganese oxides from the tailings, combined with the desorption and dissolution processes that moved the Al, Fe, Mn and other contaminants from the solid phases into the water column, accounted for about 50% of the data variability. In addition to this quantification, the results of PCA distinguished dissolved from particulate transport as two independent sources of data variation, linked to PC1 and PC2. Finally, the "anomalous" station was clearly distinguished as a site where the river water interacted with the silts and clays of the B1 dam tailings, releasing Fe and Mn into the aqueous phase, a result not verified for the "upstream" station. Put another way, the PCA revealed the fingerprint left by the B1 dam break in the Paraopeba River sediment and water chemistry.

The machine learning methods showed high accuracy and precision, with emphasis on the random forest regressor, which showed the best accuracy (RMSE less than 0.2) regardless of the target (dissolved and total concentrations of Al, Fe and Mn) and the best precision (average $R^2_{Adjust}$ higher than 90%).

Considering the accurate results obtained with the machine learning methods for the Paraopeba River, which describe the river system after the collapse of the B1 tailings dam that released an immense amount of harmful metals, a predictive platform was developed to help answer questions such as: (i) to estimate of the moment when the "anomalous" station will be in a state identical to the current state of PT-52, i.e., the return time to a pre-rupture state; (ii) to estimate the moment when the "natural" PT-19 station will have a contamination level similar to the current state of the "anomalous" station. In addition to modeling the different river segments, climate projections will be taken into account in future works, considering the evolution of atmospheric concentrations of greenhouse gases and other radiative forcings, as well as socioeconomic parameters.

As final remark, we must say that the aim of this study was to put environmental modeling at the service of scientific, technical and political actors, providing a tool to model contaminants in complex river systems, namely, those profoundly disturbed by the sudden incorporation of huge amounts of mine tailings, with the ultimate goal of restoring a vital resource—WATER— and delivering it back to the benefit of Brumadinho and other local communities.

The present study has demonstrated the ability of machine learning methods to process a considerable amount of data and produce very good predictive results. There are also a variety of deep learning algorithms that can be applied to this type of spatio-temporal data, such as Long Short Term Memory Networks (LSTMs).

**Supplementary Materials:** The following supporting information can be downloaded at: https://www.mdpi.com/article/10.3390/w16030379/s1, Datafile: Supplementary Materials_4 Stations.xlsx (datasheets: Measured parameters_water; Measured parameters_Sediments; PT-52_Dry; PT-13_Dry; PT-14_Dry; PT-19_Dry; PT-52_Rainy; PT-13_Rainy; PT-14_Rainy; PT-19_Rainy).

**Author Contributions:** Conceptualization, methodology, validation, Formal analysis and visualization, J.P.M. and G.d.S.R.; Investigation, resources, data curation, J.P.M., F.A.L.P., R.F.d.V.J., M.M.A.P.d.M.S., T.C.T.P., M.C.d.M., C.A.V., L.F.S.F. and G.d.S.R.; Writing—original draft preparation, review and editing, J.P.M., G.d.S.R. and F.A.L.P.; Supervision, F.A.L.P. and L.F.S.F.; Project administration, T.C.T.P. and M.C.d.M.; Funding acquisition, M.M.A.P.d.M.S. and C.A.V. All authors have read and agreed to the published version of the manuscript.

**Funding:** This study was funded by the contract n°5500074952/5500074950/5500074953, signed between the Vale S.A. company and the following research institutions: Fundação de Apoio Universitário; Universidade de Trás-Os-Montes e Alto Douro; and the Fundação Para o Desenvolvimento da Universidade Estadual Paulista Júlio de Mesquita Filho. For the authors integrated in the CITAB research centre, this work was further supported by National Funds of FCT—Portuguese Foundation for Science and Technology, under the project UIDB/04033/2020 (https://doi.org/10.54499/UIDB/04033/2020). The author integrated in the CITAB research centre is also integrated in the Inov4Agro—Institute for Innovation, Capacity Building and Sustainability of Agri-food Production. The Inov4Agro is an Associate Laboratory composed of two R&D units (CITAB & GreenUPorto). For the author integrated in the CQVR, the research was additionally supported by National Funds of FCT—Portuguese Foundation for Science and Technology, under the projects UIDB/00616/2020 and UIDP/00616/2020.

**Data Availability Statement:** Data are contained within the article (Supplementary Materials).

**Acknowledgments:** The authors thank Vale S.A. company, Brazil, for access to data.

**Conflicts of Interest:** The authors declare no conflicts of interest. The funders had no role in the design of the study; in the collection, analyses, or interpretation of data; in the writing of the manuscript, or in the decision to publish the results.

## Abbreviations

| | |
|---|---|
| Al | Aluminum |
| Al(dis) | Aluminum dissolved |
| Al(sed) | Aluminum sediments |
| Al(tot) | Aluminum total |
| As(dis) | Arsenic dissolved |
| As(sed) | Arsenic sediments |
| As(tot) | Arsenic total |
| B1 | mine-tailings dam (Córrego do Feijão mine of Vale, S.A) |
| BCF-RL-yy | hydrometric station (n° yy) |
| DO | Dissolved oxygen |
| Eh | Redox potential |
| Fe | Iron |
| Fe(dis) | Iron dissolved |
| Fe(sed) | Iron sediments |
| Fe(tot) | Iron total |
| m.a.s.l. | metres above sea level |
| MLP | Multilayer perceptron (neural network model) |
| MLR | Multiple linear regression (artificial intelligence model) |
| Mn | Manganese |
| Mn(dis) | Manganese dissolved |
| Mn(sed) | Manganese sediments |

| Mn(tot) | Manganese total |
|---|---|
| P(dis) | Phosphorus dissolved |
| P(sed) | Phosphorus sediments |
| P(tot) | Phosphorus total |
| Pb(dis) | Lead dissolved |
| Pb(sed) | Lead sediments |
| Pb(tot) | Lead total |
| PCA | Principal Component Analysis |
| PC1 | Chemical Characteristics (PCA result) |
| PC2 | Granulometric Characteristics (PCA result) |
| PT-xx | Monitoring station (n° xx) |
| $R^2$ | Coefficient of determination |
| RF | Random Forest (regressor models based on decision trees/machine learning algorithm) |
| RMSE | Root Mean Squared Error (accuracy) |
| sandC | Coarse-grained sand (0.500–1.000 mm) |
| sandF | Fine-grained sand (0.125–0.25 mm) |
| sandM | Sand (0.250–0.500 mm) |
| sandVC | Very coarse-grained sand (1.00–2.00 mm) |
| sandVF | Very fine-grained sand (0.062–0.125 mm) |
| T | Temperature |
| Tb | Turbidity |

## Appendix A. Results of Multiple Linear Regression with Stepwise forward Selection of Variables to Estimate Concentrations of Aluminum, Iron and Manganese

**Table A1.** Multiple linear regression with stepwise forward selection of variables, for the estimation of dissolved (Al(dis)) and total (Al(tot)) aluminum concentrations. The precision was assessed by the $R^2_{Adjust}$ (Equation (1)). The accuracy values measured by the RMSE (Equation (2)) were less than 2 for Al(tot) and less than 0.2 for Al(dis).

| Station/Period | Target | Features/Standardized Coefficients | | | | | | | | | | | $R^2_{Adjust}$ |
|---|---|---|---|---|---|---|---|---|---|---|---|---|---|
| "Upstream" | Al(dis) | Const 0.186 | Al(tot) 0.15 | Pb(dis) −0.032 | Pb(tot) 0.029 | Fe(dis) 0.071 | Fe(tot) −0.127 | P(dis) 0.031 | Mn(dis) 0.008 | Turb 0.025 | sandM 0.001 | sandVC −0.017 | 0.77 |
| "anomalous" PT-13 | Al(dis) | Const 0.206 | Pb(dis) −0.045 | Fe(dis) 0.043 | Mn(dis) 0.041 | Mn(tot) 0.059 | pH −0.033 | T 0.015 | Turb 0.053 | Al(sed) −0.037 | As(sed) 0.019 | P(sed) −0.026 | 0.65 |
| "Upstream" Dry | Al(dis) | Const 0.099 | Al(tot) 0.071 | As(tot) −0.003 | Pb(tot) 0.014 | Fe(dis) 0.057 | Fe(tot) −0.091 | Mn(tot) 0.036 | SandVF 0.004 | SandM 0.016 | SandC −0.012 | Q −0.015 | 0.87 |
| "Upstream" Rainy | Al(dis) | Const 0.271 | Al(tot) 0.159 | Pb(dis) −0.035 | Pb(tot) 0.026 | Fe(dis) 0.05 | Fe(tot) −0.011 | P(dis) 0.052 | P(tot) −0.024 | Mn(dis) 0.014 | T −0.026 | SandVC −0.011 | 0.64 |
| "anomalous" PT-13 Dry | Al(dis) | Const 0.095 | Al(tot) 0.048 | Pb(tot) −0.025 | Mn(tot) −0.016 | Turb 0.011 | Pb(sed) −0.003 | P(sed) −0.028 | Mn(sed) 0.022 | SandVF −0.013 | SandC −0.009 | Qmed 0.006 | 0.84 |
| "anomalous" PT-13 Rainy | Al(dis) | Const 0.321 | Al(tot) 0.031 | Pb(dis) −0.019 | Fe(dis) 0.163 | pH −0.02 | T −0.042 | Al(sed) −0.021 | Pb(sed) −0.041 | Fe(sed) 0.023 | SandC −0.042 | Q −0.064 | 0.73 |
| "Upstream" | Al(tot) | Const 3.011 | Al(dis) 0.458 | As(tot) 0.148 | Pb(dis) 0.276 | Pb(tot) −0.379 | Fe(dis) −0.579 | Fe(tot) 3.895 | P(dis) 0.21 | Mn(tot) −0.428 | P(sed) −0.378 | Mn(sed) 0.335 | 0.97 |
| "anomalous" PT-13 | Al(tot) | Const 3.164 | As(tot) 1.028 | Fe(tot) 2.675 | P(tot) 0.386 | Mn(dis) −0.968 | Mn(tot) −1.801 | pH −0.245 | Turb 2.138 | Al(sed) −0.209 | P(sed) −0.154 | SandF −0.198 | 0.90 |
| "Upstream" Dry | Al(tot) | Const 0.558 | Al(dis) 0.127 | As(tot) −0.041 | Pb(tot) 0.091 | Fe(dis) −0.214 | Fe(tot) 0.379 | P(dis) 0.030 | pH −0.041 | Turb 0.151 | SandF 0.033 | Q −0.004 | 0.93 |
| "Upstream" Rainy | Al(tot) | Const 5.402 | Al(dis) 0.53 | As(tot) 0.228 | Pb(dis) 0.347 | Pb(tot) −0.391 | Fe(dis) −0.465 | Fe(tot) 3.574 | P(dis) 0.34 | Mn(tot) −0.42 | T 0.198 | Turb 0.291 | 0.95 |
| "anomalous" PT-13 Dry | Al(tot) | Const 0.569 | Al(dis) 0.198 | As(tot) 0.064 | Pb(tot) 0.077 | Fe(dis) 0.265 | Fe(tot) 0.289 | P(tot) 0.105 | Mn(tot) 0.043 | Turb −0.198 | P(sed) 0.112 | Mn(sed) −0.072 | 0.92 |
| "anomalous" PT-13 Rainy | Al(tot) | Const 5.861 | As(tot) 1.388 | Fe(tot) 3.643 | P(dis) 0.621 | Mn(dis) −1.142 | Mn(tot) −2.736 | pH −0.341 | Turb 2.008 | Al(sed) −0.526 | Mn(sed) −0.821 | Silt 0.711 | 0.87 |

**Table A2.** Multiple linear regression with stepwise forward selection of variables, for the estimation of dissolved (Fe(dis)) and total (Fe(tot)) iron concentrations. The precision was assessed by the $R^2_{Adjust}$ (Equation (1)). The accuracy values measured by the RMSE (Equation (2)) were less than 2 for Fe(tot) and less than 0.2 for Fe(dis).

| Station/Period | Target | Features/Standardized Coefficients | | | | | | | | | | | $R^2_{Adjust}$ |
|---|---|---|---|---|---|---|---|---|---|---|---|---|---|
| "Upstream" | Fe(dis) | Const 0.437 | Al(dis) 0.084 | Al(tot) −0.216 | Pb(dis) 0.034 | Fe(tot) 0.241 | P(dis) 0.088 | pH −0.015 | T 0.022 | Turb −0.069 | Mn(sed) −0.014 | SandM 0.031 | 0.80 |
| "anomalous" PT-13 | Fe(dis) | Const 0.692 | Al(dis) 0.196 | Fe(tot) −0.375 | P(dis) −0.111 | Mn(dis) 0.368 | Turb 0.395 | As(sed) −0.103 | Pb(sed) −0.136 | P(sed) 0.243 | Silt 0.086 | SandVF −0.102 | 0.56 |
| "Upstream" Dry | Fe(dis) | Const 0.336 | Al(dis) 0.061 | Al(tot) −0.109 | Pb(dis) 0.032 | Pd(tot) −0.015 | Fe(tot) 0.162 | P(dis) 0.010 | Mn(tot) −0.049 | pH −0.007 | SandF 0.013 | Q 0.063 | 0.93 |
| "Upstream" Rainy | Fe(dis) | Const 0.536 | Al(dis) 0.05 | Al(tot) −0.105 | As(tot) 0.035 | Pb(dis) 0.045 | Fe(tot) 0.078 | P(dis) 0.097 | Turb −0.063 | As(sed) 0.055 | Mn(sed) −0.089 | As(sed) 0.026 | 0.78 |
| "anomalous" PT-13 Dry | Fe(dis) | Const 0.586 | Al(dis) −0.004 | Al(tot) 0.331 | As(tot) 0.192 | Fe(tot) −0.477 | P(dis) −0.243 | Mn(tot) 0.155 | Turb 0.283 | Fe(sed) −0.111 | Mn(sed) 0.087 | Q 0.090 | 0.93 |
| "anomalous" PT-13 Rainy | Fe(dis) | Const 0.803 | Al(dis) 0.212 | Al(tot) −0.117 | Pd(tot) −0.082 | Fe(tot) 0.264 | Mn(dis) 0.409 | Mn(tot) −0.144 | Pb(sed) 0.056 | Mn(sed) −0.036 | SandF −0.026 | Q 0.11 | 0.88 |
| "Upstream" | Fe(tot) | Const 4.078 | Al(dis) −0.311 | Al(tot) 2.605 | Pb(dis) −0.086 | Pd(tot) 0.486 | Fe(dis) 0.449 | P(dis) −0.177 | Mn(dis) −0.071 | Mn(tot) 0.721 | T 0.074 | Turb 0.283 | 0.98 |
| "anomalous" PT-13 | Fe(tot) | Const 4.986 | Al(tot) 1.881 | As(tot) 0.494 | Fe(dis) −0.571 | P(tot) −0.601 | Mn(tot) 4.846 | pH −0.215 | Turb 1.442 | Pb(sed) −0.255 | SandVF −0.766 | SandF 0.334 | 0.96 |
| "Upstream" Dry | Fe(tot) | Const 1.315 | Al(dis) −0.244 | Al(tot) 0.326 | As(tot) 0.043 | Fe(dis) 0.327 | P(tot) 0.028 | Mn(dis) −0.046 | Mn(tot) 0.376 | Turb 0.204 | As(sed) 0.023 | Q −0.19 | 0.98 |
| "Upstream" Rainy | Fe(tot) | Const 6.771 | Al(dis) −0.266 | Al(tot) 2.684 | Pd(tot) 0.439 | Fe(dis) 0.316 | P(tot) −0.121 | Mn(tot) 0.795 | Turb 0.328 | P(sed) 0.484 | Mn(sed) −0.781 | SandC 0.131 | 0.97 |
| "anomalous" PT-13 Dry | Fe(tot) | Const 1.28 | Al(tot) 0.717 | As(tot) −0.174 | Fe(dis) −1.085 | P(dis) −0.542 | Mn(dis) −0.114 | Mn(tot) 0.472 | Turb 0.673 | Al(sed) 0.058 | As(sed)g −0.135 | Q 0.156 | 0.91 |
| "anomalous" PT-13 Rainy | Fe(tot) | Const 8.838 | Al(tot) 2.135 | As(tot) 0.485 | Fe(dis) 0.908 | P(dis) −0.676 | Mn(dis) −0.763 | Mn(tot) 6.168 | Turb 1.437 | As(sed)g 0.327 | SandVF −1.573 | SandF 0.984 | 0.96 |

**Table A3.** Multiple linear regression with stepwise forward selection of variables, for the estimation of dissolved (Mn(dis)) and total (Mn(tot)) manganese concentrations. The precision was assessed by the $R^2_{Adjust}$ (Equation (1)). The accuracy values measured by the RMSE (Equation (2)) were less than 2 for Mn(tot) and less than 0.2 for Mn(dis).

| Station/Period | Target | Features/Standardized Coefficients | | | | | | | | | | | $R^2_{Adjust}$ |
|---|---|---|---|---|---|---|---|---|---|---|---|---|---|
| "Upstream" | Mn(dis) | Const 0.022 | Al(dis) 0.002 | Fe(dis) −0.002 | P(tot) 0.005 | Al(sed) −0.002 | Pb(sed) 0.002 | Fe(sed) −0.003 | As(sed) −0.003 | SandF −0.002 | SandC −0.004 | SandVF 0.001 | 0.37 |
| "anomalous" PT-13 | Mn(dis) | Const 0.068 | Al(dis) 0.006 | Al(tot) −0.041 | As(tot) 0.026 | Fe(dis) 0.021 | P(tot) −0.020 | Mn(tot) 0.127 | As(sed) 0.008 | Silt −0.009 | SandVF 0.011 | Q −0.01 | 0.87 |
| "Upstream" Dry | Mn(dis) | Const 0.017 | Pb(dis) 0.004 | Fe(dis) −0.003 | Fe(tot) −0.006 | P(tot) 0.002 | Mn(tot) 0.008 | pH −0.002 | As(sed) −0.002 | Silt 0.005 | Sand_m 0.005 | Q −0.003 | 0.61 |
| "Upstream" Rainy | Mn(dis) | Const 0.025 | Al(dis) 0.003 | As(tot) 0.003 | Pd(tot) −0.002 | P(tot) 0.006 | T 0.001 | Fe(sed) −0.005 | As(sed) −0.004 | SandVF 0.001 | SandC −0.005 | SandVF 0.003 | 0.31 |
| "anomalous" PT-13 Dry | Mn(dis) | Const 0.042 | As(tot) 0.015 | Pb(dis) 0.016 | Pd(tot) −0.020 | Mn(tot) 0.003 | T 0.005 | Pb(sed) 0.008 | P(sed) −0.009 | Mn(sed) 0.010 | Silt 0.003 | SandVF 0.002 | 0.72 |
| "anomalous" PT-13 Rainy | Mn(dis) | Const 0.095 | Al(dis) −0.034 | As(tot) 0.021 | Pd(tot) 0.023 | Fe(dis) 0.111 | Fe(tot) −0.058 | P(dis) −0.023 | Mn(tot) 0.139 | Pb(sed) −0.014 | SandVF 0.007 | Q −0.029 | 0.91 |
| "Upstream" | Mn(tot) | Const 0.383 | Al(dis) 0.029 | Al(tot) −0.161 | As(tot) 0.069 | Pb(dis) −0.030 | Pd(tot) −0.055 | Fe(tot) 0.437 | P(dis) 0.023 | P(tot) 0.061 | Mn(dis) 0.016 | As(sed) 0.014 | 0.89 |
| "anomalous" PT-13 | Mn(tot) | Const 0.818 | Al(tot) −0.291 | As(tot) −0.127 | Pb(dis) 0.068 | Fe(dis) 0.067 | Fe(tot) 1.161 | P(tot) 0.228 | Mn(dis) 0.448 | pH 0.071 | SandVF 0.164 | SandF −0.129 | 0.96 |
| "Upstream" Dry | Mn(tot) | Const 0.121 | Al(dis) 0.026 | Pd(tot) −0.013 | Fe(dis) −0.028 | Fe(tot) 0.084 | P(tot) −0.007 | Mn(dis) 0.013 | pH 0.003 | Turb −0.034 | SandVC 0.005 | Q 0.034 | 0.94 |
| "Upstream" Rainy | Mn(tot) | Const 0.637 | Al(tot) −0.219 | As(tot) 0.084 | Pb(dis) −0.040 | Pd(tot) −0.052 | Fe(tot) 0.473 | P(dis) 0.053 | P(tot) 0.059 | As(sed) 0.039 | P(sed) −0.123 | Mn(sed) 0.096 | 0.80 |
| "anomalous" PT-13 Dry | Mn(tot) | Const 0.213 | Al(dis) −0.023 | Al(tot) 0.014 | Fe(dis) 0.057 | Fe(tot) 0.087 | P(dis) 0.076 | Mn(dis) 0.031 | Turb 0.020 | Pb(sed) −0.015 | Fe(sed) 0.013 | Q −0.017 | 0.87 |
| "anomalous" PT-13 Rainy | Mn(tot) | Const 1.446 | Al(tot) −0.395 | As(tot) −0.148 | Pb(dis) 0.140 | Fe(tot) 1.516 | P(dis) 0.222 | Mn(dis) 0.549 | As(sed)g −0.089 | SandVF 0.321 | SandF −0.289 | SandVC −0.089 | 0.96 |

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
