# Peer review of "The Modeling of a River Impacted with Tailings Mudflows Based on the Differentiation of Spatiotemporal Domains and Assessment of Water–Sediment Interactions Using Machine Learning Approaches"

_water, doi:10.3390/w16030379_

Round 1

Reviewer 1 Report

Comments and Suggestions for Authors

1.       The authors repeatedly refer to ‘anomalous PT-13 stations’ throughout the text. It is suggested that they find out a short or contracted form for this phrase as this would improve the text

2.       Line 556,    correct ‘comportment’ spelling

3.       Line 714-715, the authors reported lesser accuracy for the MLP NN than for the MLR. This is a surprising result, as ML methods are more robust in capturing the non-linear interactions among the variables. The authors need to justify this result.

4.       The measured concentrations at the stations would have been divided into ‘training’ and ‘validating’ dataset. The authors need to provide this information for the benefit of the prospective research.

5.       The authors have two stations in the anomalous zone, i.e. PT 13 and PT 14 but they have only talked about PT-13 only. What may be the reason for it? The narration would be more complete if they advance a reason for this.

Author Response

Response to Reviewer 1

First, we would like to thank the reviewer for his comments and suggestions. In this document, we provide a response to each of the comments in the sections below. We are confident that the reviewer's comments and suggested changes will improve the quality of the article.

The blue text is the reviewer's comment/suggestion.

Author response: The black text is the response to the comment/suggestion.

Line 999. The text highlighted in yellow is the text that has been added or changed in the paper.

  1. The authors repeatedly refer to ‘anomalous PT-13 stations’ throughout the text. It is suggested that they find out a short or contracted form for this phrase as this would improve the text.

Author response: We agreed with the reviewer's suggestion and changed ‘anomalous PT-13 station’ to ‘anomalous station’ throughout the text.

Line 216. It also shows that the “anomalous” segment is located between of the confluence of Ribeirão Ferro-Carvão with the Paraopeba River and the physical barrier called Igarapé weir (PT-14, located 53.9 km downstream of B1 dam site). The PT-13 station, hereafter referred to as the “anomalous station”, corresponds to the sector most impacted by the tailings dump, as per the report of Arcadis company about mechanical drillings executed along the river and corresponding characterization of sediment and tailings testimonies [45].

  1. Line 556, correct ‘comportment’ spelling

Author response: ‘comportment’ is correct.

Line 560. Thus, the non-clay fractions maintained a non-cohesive behavior, responding to flow in-creases the same way as in the dry period, while the clay fraction’s comportment reversed due to cohesive forces. Regardless the season, the resuspension of sediments was accompanied by the mobilization of metals.

  1. Line 714-715, the authors reported lesser accuracy for the MLP NN than for the MLR. This is a surprising result, as ML methods are more robust in capturing the non-linear interactions among the variables. The authors need to justify this result.

Author response: In reality, the 2 models give similar results. It should be noted that the data used come from an area affected by an environmental disaster and that the Paraopeba River is a natural, open system with a lot of variability, for example in the dry and rainy periods. MLP NN performs best in the following sections.

Line 720. The fit of MLP neural networks resulted in R2adjust = 0.79 (on average), which is similar than that of MLR models (0.84).

  1. The measured concentrations at the stations would have been divided into ‘training’ and ‘validating’ dataset. The authors need to provide this information for the benefit of the prospective research.

Author response: The data used corresponds to the period from the dam failure in 2019 to the end of 2021 at the time of this study. Thus, the training method was cross-validation with 3 folds.

Line 342. The training method was identical for all models and cross-validation with 3 folds was used.

  1. The authors have two stations in the anomalous zone, i.e. PT 13 and PT 14 but they have only talked about PT-13 only. What may be the reason for it? The narration would be more complete if they advance a reason for this.

Author response: We appreciate the suggestion and have improved the text to make it clearer (modified sentences are highlighted in yellow).

Line 216. It also shows that the “anomalous” segment is located between of the confluence of Ribeirão Ferro-Carvão with the Paraopeba River and the physical barrier called Igarapé weir (PT-14, located 53.9 km downstream of B1 dam site). The PT-13 station, hereafter referred to as the “anomalous station”, corresponds to the sector most impacted by the tailings dump, as per the report of Arcadis company about mechanical drillings executed along the river and corresponding characterization of sediment and tailings testimonies [45].

Lines 445, 450, 452, 468, 471, 538, 593, 596, 646, 658, 659, 824, 830, 848, 860, 863.

Reviewer 2 Report

Comments and Suggestions for Authors

The authors effectively use statistical models to analyze the data of the impact of the dam collapse on the Paraopeba River. Overall, the methodology of the study is sound, but there are areas where the authors could provide more detail or clarification. 

Title needs be amended (or re-written) because the term “river model” alone does not capture what is being modelled. It is suggested to be explicit about what is being modelled.

The dataset used in the study includes the physical-chemical characterization of 270 parameters at 65 different locations. For this study, only the values of 30 parameters were used - can the authors offer more context on the selection process of the parameters? What is the rationale for the selection of these 30 parameters out of the 270 available?

Can the authors comment on the instrumentation used to capture the data used in the study and methods used to offer estimates for these data? Are there any assumptions used?

Can the authors clarify in their main text the distinction of accuracy vs precision? 

Given there is an amount of error on the original data, the exact modelling of these data (by means of stochastic or numerical methods) is not going to render them accurate, right? For this reason the “true” values should be available, while only “measured” values are accessible. I am interested in seeing the authors clearly elaborating on this and accepting any limitations of the used models (which may be precise, but one can claim accuracy only under the assumption of knowledge of the true values).

The experimental phase of the study involved the use of statistical models to assess water-sediment interactions. The models were fed with the dataset and the spatial and temporal domains defined in the study. The analyses were applied to the "upstream" and "anomalous" PT-13 segments of the Paraopeba River, which were the most contrasting of natural and nonnatural sectors of the river. Can the authors be clear what they refer to as “non-natural” sectors and how can they be certain that these “anomalies” aren’t attributed to processes the authors perhaps have not considered? It will be interesting to the reader to understand more about the assumptions made herein and any processes normally considered (or not).

The experimental phase of the study involved the use of statistical models to assess water-sediment interactions. The authors used Spearman rank-order correlation and Principal Component Analysis, which are appropriate for this type of study. However, the authors could provide more details about the model selection process and why these specific models were chosen. Additionally, the authors could provide more information about the assumptions made in these models and how they validated these assumptions. This would strengthen the study by providing more transparency about the methods used and the robustness of the findings.

The study concluded that the main contribution and novelty of the work is the combination of the segmentation of a river system to allow the definition of homogeneous sectors capable of being interpretable from geochemical and environmental standpoints, with the use of artificial intelligence approaches capable of connecting the immense number and diversity of variables that contribute to the river and sediment water quality. Can the authors add a paragraph where they can present a generalised framework of the method they have applied which may help practitioners to implement it to other similar case studies? 

What are the specific assumptions and limitations over the types and amount of data that are needed in order to allow one confidently apply the same approach to their own region?

Could the authors provide more context on the implications of their findings for the local population and environment? For example, they could discuss the potential health risks associated with exposure to the heavy metals found in the water and sediments of the river. Additionally, the authors could outline some potential strategies for mitigating the impact of the dam collapse on the river and its ecosystem, following the findings of their study, in the discussion section. For example, how can this study’s results benefit the decision making process and management of the river towards mitigation of the dam collapse impacts?

I am interested also to see a couple of lines of the author’s perspective of the future of such AI methods and if they would envisage using some other tools next.

Comments on the Quality of English Language

Ok overall.

Here are some examples from the manuscript where the writing can be improved, along with suggestions for improvement:

1. Original: "Artificial intelligence models were trained in the second phase of Figure 2's workflow, called "modeling phase”.”

   Suggestion: "In the second phase of the workflow depicted in Figure 2, known as the "modeling phase", artificial intelligence models were trained."

2. Original: "Regarding the "upstream" station in the dry period (Figure 9a), the results suggested principal component PC1 to be called "Chemical Characteristics", because the chemistry of sediments (compartment (E); Table 1) have clustered together in opposition to the total concentrations of metals in water (compartment (B))."

   Suggestion: "For the "upstream" station in the dry period (Figure 9a), the results suggested that the principal component PC1 should be termed "Chemical Characteristics", as the chemistry of the sediments (compartment (E); Table 1) clustered together, contrasting with the total concentrations of metals in water (compartment (B))."

3. Original: "In the present study, they comprised multiple linear regression models with stepwise forward selection of variables, multilayer perceptron neural networks, and random forest regressor."

   Suggestion: "In this study, the models used included multiple linear regression models with stepwise forward selection of variables, multilayer perceptron neural networks, and random forest regressors."

Author Response

Response to Reviewer 2

First, we would like to thank the reviewer for his comments and suggestions. In this document, we provide a response to each of the comments in the sections below. We are confident that the reviewer's comments and suggested changes will improve the quality of the article.

The blue text is the reviewer's comment/suggestion.

Author response: The black text is the response to the comment/suggestion.

Line 999. The text highlighted in yellow is the text that has been added or changed in the paper.

Title needs be amended (or re-written) because the term “river model” alone does not capture what is being modelled. It is suggested to be explicit about what is being modelled.

Author response: Thank you for your comment and change the title to:

Modeling of an impacted river based on the differentiation of spatiotemporal domains and assessment of water-sediment interactions using machine learning approaches

The dataset used in the study includes the physical-chemical characterization of 270 parameters at 65 different locations. For this study, only the values of 30 parameters were used - can the authors offer more context on the selection process of the parameters? What is the rationale for the selection of these 30 parameters out of the 270 available?

Author response:   Vale S.A. company, the owner of the tailings dam, has been ordered by a court to monitor a series of parameters along the Paraopeba River, from the Brumadinho dam to the Atlantic Ocean, in order to prevent the damage from spreading.

After the collapse of the dam, the Paraopeba River was divided into 3 segments: the non-impacted, the impacted and the potentially impacted. Station PT-52 is located 12.6 km upstream of dam B1 (non-impacted area). The impacted segment is located between stations PT-13 and PT-14 (PT-14 is located 53.9 km downstream of dam B1). The PT-13 station corresponds to the sector most impacted by the tailings.  Station PT-19 is located 249.8 km downstream of dam B1 and corresponds to a sector not yet affected by the dam collapse.

Vale S.A. company is required by court order to collect a large amount of data, for example to determine the presence of hydrocarbons in the water. Based on our experience from several studies already carried out, we selected the data suitable for this study. The selected data are directly related to river water quality, contaminants concentrations (dissolved and total), sediment chemical composition, sediment grain size fractions and streamflow (Table 1), collected at stations PT-52, PT-13, PT-14 and PT-19 (Figure 1).

Can the authors comment on the instrumentation used to capture the data used in the study and methods used to offer estimates for these data? Are there any assumptions used?

Author response:  We have quality assurance and control procedures described in detail in our article published in the journal Applied Geochemistry and entitled "Geochemistry and contamination of sediments and water in rivers affected by the collapse of tailings dams (Brumadinho, Brazil)", section 2.3.4 (https://doi.org/10.1016/j.apgeochem.2023.105644).

In summary, Sediment sampling followed the ABNT NBR 10.007:2004 standard, particularly with regard to the use of appropriate sampling equipment and the correct preservation of samples. The sampling of water followed international guidelines, which include the same precautions as for sediment sampling, supplemented by the use of calibrated equipment for the measurement of physico-chemical parameters and the correct preservation of samples. Duplicate samples were taken to check the accuracy and repeatability of the procedure.  Water sampling was checked against the temperature blank and against equipment blanks to verify that the cleaning of the equipment between sampling points was effective. The analysis phase of the water samples included the QA/QC procedures of Matrix Spike and Matrix Spike Duplicate, used to evaluate the recovery and accuracy of the method; limits of quantification with reference to the maximum allowable values of the Brazilian Resolution CONAMA 357/2005 and the Joint Normative Resolution COPAM/CERH-MG no. January 2008; method blank; laboratory control samples. The results of agreement were 99.7% for the matrix spike tests, 100% for the method blank, 97.6% for the laboratory control samples and 100% for the quantification limits and surrogates.

Can the authors clarify in their main text the distinction of accuracy vs precision?

Author response: We have included a short explanation of the accuracy (RMSE) and precision (R2adjust) in the text.

Line 348. R2adjust is the percentage of the model's ability to correctly estimate the dependent variable as a function of the independent variables. The closer it is to 1, the better the model.

Line 355. RMSE is the average error between the predicted and observed values in a data set. The smaller the value, the smaller the error.

Given there is an amount of error on the original data, the exact modelling of these data (by means of stochastic or numerical methods) is not going to render them accurate, right? For this reason the “true” values should be available, while only “measured” values are accessible. I am interested in seeing the authors clearly elaborating on this and accepting any limitations of the used models (which may be precise, but one can claim accuracy only under the assumption of knowledge of the true values).

Author response:  We understand that the variables measured by the sensors in the Paraopeba river basin are considered as reference values. The sensors have their accuracy specifications and confidence intervals, which can be checked in the references listed in the section 2.2.1. Measurement protocols.

In this study, when we present the RMSE and R2adjust values, they correspond to the measures of accuracy and precision, respectively. Accuracy is the general measure of goodness of fit, the estimated values relative to the observed values. It includes random and systematic error. Precision refers to the repeatability of the estimated values. Understands only systematic error. One of the problems we encounter when training models is the time scale to be considered. We received weekly data with high variability, but the machine learning models managed to achieve high accuracy in their adjustments. The calibration of mechanistic models would be more difficult.

The experimental phase of the study involved the use of statistical models to assess water-sediment interactions. The models were fed with the dataset and the spatial and temporal domains defined in the study. The analyses were applied to the "upstream" and "anomalous" PT-13 segments of the Paraopeba River, which were the most contrasting of natural and nonnatural sectors of the river. Can the authors be clear what they refer to as “non-natural” sectors and how can they be certain that these “anomalies” aren’t attributed to processes the authors perhaps have not considered? It will be interesting to the reader to understand more about the assumptions made herein and any processes normally considered (or not).

Author response: Thank you for your suggestion to clarify the text.

Line 302. In the present study, we applied these and subsequent analyses only to the “upstream” and “anomalous” PT-13 segments, because they were the most contrasting of non-impacted and impacted sectors of Paraopeba River in terms of their relationship to the B1 dam collapse, and hence the best to differentiate water-sediment interactions ac-cording to sediment source (natural sediments or natural sediments + mine tailings, respectively).

The experimental phase of the study involved the use of statistical models to assess water-sediment interactions. The authors used Spearman rank-order correlation and Principal Component Analysis, which are appropriate for this type of study. However, the authors could provide more details about the model selection process and why these specific models were chosen. Additionally, the authors could provide more information about the assumptions made in these models and how they validated these assumptions. This would strengthen the study by providing more transparency about the methods used and the robustness of the findings.

Author response: The choice of machine learning models is due to the fact that they do not make any theoretical assumptions about the behaviour of the phenomenon under study. Thus, the relationship between system variables can be freely captured. Machine learning models are not specific to one phenomenon (mechanistic or procedural), the same model can be used for different purposes. At the same time, machine learning models have a high ability to accurately estimate or predict values, making them very useful for strategic decisions, such as in dams.

The study concluded that the main contribution and novelty of the work is the combination of the segmentation of a river system to allow the definition of homogeneous sectors capable of being interpretable from geochemical and environmental standpoints, with the use of artificial intelligence approaches capable of connecting the immense number and diversity of variables that contribute to the river and sediment water quality. Can the authors add a paragraph where they can present a generalised framework of the method they have applied which may help practitioners to implement it to other similar case studies?

Author response: Thank you for your suggestion to clarify the text.

Line 136. Taken all these insights and potential actions together, the main contribution and novelty of this work lies in combining, in a single study, the segmentation of a river system that was divided into 3 segments due to a technological accident, not impacted, impacted and potentially impacted. Each homogeneous sector will be analysed from a geochemical and environmental point of view, using artificial intelligence approaches capable of linking the immense number and diversity of variables that contribute to the quality of water in rivers and sediments and, most importantly, building a predictive structure capable of anticipating the river’s future.

What are the specific assumptions and limitations over the types and amount of data that are needed in order to allow one confidently apply the same approach to their own region?

Author response: In order to provide a better response to this suggestion, we have added a new section 3.4 Limitations, implications and future work.

Line 789. The methodology presented has 3 main limitations: (1) it applies to rivers that have suffered a technological accident, resulting in the division of the river into 3 distinct segments. We will have a non-impacted segment, an impacted segment and finally a potentially impacted segment. (2) The wastes are metals with water-sediment interactions. (3) The climate is tropical with a strong contrast in rainfall between the rainy and dry periods.

Could the authors provide more context on the implications of their findings for the local population and environment? For example, they could discuss the potential health risks associated with exposure to the heavy metals found in the water and sediments of the river. Additionally, the authors could outline some potential strategies for mitigating the impact of the dam collapse on the river and its ecosystem, following the findings of their study, in the discussion section. For example, how can this study’s results benefit the decision making process and management of the river towards mitigation of the dam collapse impacts?

Author response: In order to provide a better response to this suggestion, we have added a new section 3.4 Limitations, implications and future work.

Line 794. This article is part of a project called Entire which aims to study the impact of the tailings from the Brumadinho dam collapse for a restored aquatic environment and aims to model the 3 segments of the river to later be able to simulate the behaviour of the river as a whole. The collapse of the Brumadinho dam has prompted a wide range of studies, which have resulted in numerous publications with valuable contributions in various areas, such as the economic, social, health, fauna, environmental and aquatic ecosystems.

I am interested also to see a couple of lines of the author’s perspective of the future of such AI methods and if they would envisage using some other tools next.

Author response: We have added another topic to the conclusions on AI.

Line 873. The present study has demonstrated the ability of Machine Learning methods to process a considerable amount of data and produce very good predictive results. There are also a variety of Deep Learning algorithms that can be applied to this type of spatio-temporal data, such as Long Short Term Memory Networks (LSTMs).

Comments on the Quality of English Language Ok overall.

Author: We are very grateful for the suggestions for improvement.

Here are some examples from the manuscript where the writing can be improved, along with suggestions for improvement:

  1. Original: "Artificial intelligence models were trained in the second phase of Figure 2's workflow, called "modeling phase”.”

Suggestion: "In the second phase of the workflow depicted in Figure 2, known as the "modeling phase", artificial intelligence models were trained." Line 315.

  1. Original: "Regarding the "upstream" station in the dry period (Figure 9a), the results suggested principal component PC1 to be called "Chemical Characteristics", because the chemistry of sediments (compartment (E); Table 1) have clustered together in opposition to the total concentrations of metals in water (compartment (B))."

Suggestion: "For the "upstream" station in the dry period (Figure 9a), the results suggested that the principal component PC1 should be termed "Chemical Characteristics", as the chemistry of the sediments (compartment (E); Table 1) clustered together, contrasting with the total concentrations of metals in water (compartment (B))." Line 604.

  1. Original: "In the present study, they comprised multiple linear regression models with stepwise forward selection of variables, multilayer perceptron neural networks, and random forest regressor."

Suggestion: "In this study, the models used included multiple linear regression models with stepwise forward selection of variables, multilayer perceptron neural networks, and random forest regressors." Line 316.

Round 2

Reviewer 2 Report

Comments and Suggestions for Authors

I appreciate the author's replies.

However, I would like to see the clarifications given to me also in the main text where appropriate, as I believe many other readers may have require the same clarity and this will benefit the manuscript's presentation.

Kindly do provide these clarifications in the main text!

Comments on the Quality of English Language

ok

Author Response

Dear reviewer #2,

Please see the attached file where the answers to your comments are provided.

Best regards,

The authors
